# Unveiling and Manipulating Prompt Influence in Large Language Models

**Zijian Feng**[1,3]**, Hanzhang Zhou**[1,3]**, Zixiao Zhu**[1,3]**, Junlang Qian**[2] **& Kezhi Mao**[2,3*]
[1]Institute of Catastrophe Risk Management, Interdisciplinary Graduate Programme, Nanyang Technological University, Singapore
[2]School of Electrical and Electronic Engineering, Nanyang Technological University, Singapore
[3]Future Resilient Systems Programme, Singapore-ETH Centre, CREATE Campus, Singapore
`{feng0119, hanzhang001, zixiao001, junlang001}@e.ntu.edu.sg`
`ekzmao@ntu.edu.sg`

## Abstract

Prompts play a crucial role in guiding the responses of Large Language Models (LLMs). However, the intricate role of individual tokens in prompts, known as input saliency, in shaping the responses remains largely underexplored. Existing saliency methods either misalign with LLM generation objectives or rely heavily on linearity assumptions, leading to potential inaccuracies. To address this, we propose Token Distribution Dynamics (TDD), a simple yet effective approach to unveil and manipulate the role of prompts in generating LLM outputs. TDD leverages the robust interpreting capabilities of the language model head (LM head) to assess input saliency. It projects input tokens into the embedding space and then estimates their significance based on distribution dynamics over the vocabulary. We introduce three TDD variants: forward, backward, and bidirectional, each offering unique insights into token relevance. Extensive experiments reveal that the TDD surpasses state-of-the-art baselines with a big margin in elucidating the causal relationships between prompts and LLM outputs. Beyond mere interpretation, we apply TDD to two prompt manipulation tasks for controlled text generation: zero-shot toxic language suppression and sentiment steering. Empirical results underscore TDD's proficiency in identifying both toxic and sentimental cues in prompts, subsequently mitigating toxicity or modulating sentiment in the generated content.

## 1 Introduction

The formulation of prompts significantly shapes the textual responses of large language models (LLMs). Comprehending the influence of individual input tokens in prompts, i.e., input saliency, can augment our insight into LLM interpretability and foster the development of refined prompting strategies to modulate LLM outputs. However, input saliency in LLMs and its bearing on prompt-based generation control are rarely explored.

Although various saliency methods ranging from perturbation-based (Feng et al., 2018; Prabhakaran et al., 2019), gradient-based (Chefer et al., 2021; Ali et al., 2022), to vector-based (Modarressi et al., 2022; Ferrando et al., 2022) for Transformer-based language models have been devised to gauge input token significance, their adaptation to LLMs presents certain limitations. First, these methods are chiefly designed for text classification using masked language models such as BERT (Devlin et al., 2019) and RoBERTa (Liu et al., 2019). Their saliency explanations, limited to class labels, do not align with the objectives of autoregressive LLMs, which aim to explain token generation across the whole vocabulary. Second, many of these methods rely heavily on linearity assumptions to approximate the intricate non-linear behaviors in language models. For instance, vector-based approaches (Modarressi et al., 2022; Ferrando et al., 2022) often assume linearly combined attention weights across layers, while gradient-based ones (Wallace et al., 2019; Chefer et al., 2021; Ali et al., 2022) employ local linear approximations, referencing Taylor's expansion theorem. Such

---

[*]Corresponding author.

assumptions can lead to significant inaccuracies, especially as the number of LLM layers increases. These limitations can hinder the understanding of how LLMs respond to prompts and hence the effective use of prompts for desired outputs.

Unlike previous methods that use attention or gradient as a medium to compute saliency, in this study, we propose adopting token distributions as a means to estimate token saliency. Recent studies on GPT2 show that token representations can be visualized as distributions across the vocabulary, termed **token distributions**, through the LM head (Nostalgebraist, 2020; Geva et al., 2021; 2022; Dar et al., 2023). We expand this notion to other contemporary LLMs such as Pythia (Biderman et al., 2023) and LLaMA2 (Touvron et al., 2023), emphasizing the relationship between token distributions and token contributions. Our findings indicate that the LM head acts as an effective interpreter for LLMs, skillfully decoding ambiguous input token representations and mapping them to the embedding space. These projected token distributions within the embedding space are interpretable and comprehensible to humans, and align with model predictions. We argue that variations in token distributions across layers can be attributed to token saliency.

To unveil token saliency in prompts for LLM generation, we introduce the Token Distribution Dynamics (TDD) approach. In alignment with the generative objectives of LLMs, our methodology offers contrastive explanations (Yin & Neubig, 2022), elucidating why LLMs prioritize one token over others in the entire vocabulary. We present three variants: TDD-forward, TDD-backward, and TDD-bidirectional. TDD-forward capitalizes on token distribution dynamics throughout the progression of learning and prediction, while TDD-backward evaluates the token importance using backward token dynamics. In parallel, TDD-bidirectional integrates insights from both directional perspectives. Despite its apparent simplicity, TDD serves as an exceptionally potent tool. Comprehensive experiments show that TDD markedly outperforms advanced saliency methods in discerning the causal relationships between prompts and LLM outputs across the entire vocabulary.

Beyond merely providing interpretation, we elucidate how to harness TDD to manipulate prompts and control LLM outputs. We spotlight two key applications: zero-shot toxic language suppression and sentiment steering. In toxic language suppression, TDD identifies and neutralizes toxic triggers in prompts before they are fed into LLMs. For sentiment modulation, TDD captures sentiment cues in prompts, adjusting their sentiment inclination to guide the sentiment of generated texts. Results of extensive experiments showcase TDD's efficacy in pinpointing predefined triggers and modulating the toxicity or sentiment of the outputs.

The contribution of this study can be summarized as follows [1]:

1. We introduce a novel solution to assessing input saliency based on token distributions, which we validate and elaborate upon in autoregressive LLMs, including GPT2 (Radford et al., 2019), GPTJ (Wang & Komatsuzaki, 2021), BLOOM (Scao et al., 2022), Pythia (Biderman et al., 2023), and LLaMA2 (Touvron et al., 2023).

2. We propose TDD, a technique that leverages token distribution dynamics to unveil prompt influence in LLM generations. We present three variants that harness token dynamics from distinct directions. Extensive evaluations across various datasets and LLMs show that our techniques notably surpass other leading methods.

3. We exemplify two applications of TDD to manipulate LLM outputs, specifically targeting the reduction of toxicity in language generation and managing the sentiment of produced texts. Empirical evidence reveals that TDD effectively detects toxic or sentiment-related triggers in prompts, enabling more controlled LLM outputs.

## 2 RELATED WORK

### 2.1 INPUT SALIENCY FOR LANGUAGE MODELS

Existing saliency approaches to explaining the relationships between prompts and language model outputs can be classified into gradient-based, vector-based, and perturbation-based methods. Gradient-based methods employ backpropagation gradients to determine the significance of each token. Examples of this approach include gradient norm (Simonyan et al., 2014; Li et al., 2016a;

---

[1]Code will be released here: `https://github.com/zijian678/TDD`

Atanasova et al., 2020), gradient $\times$ input (Denil et al., 2014; Shrikumar et al., 2017; Wallace et al., 2019), LRP-XAI (Bach et al., 2015; Ali et al., 2022), and generic attention explainability (GAE) (Chefer et al., 2021). For vector-based methods, rather than utilizing the last layer's attention weights (Bahdanau et al., 2014), techniques like attention rollout (Abnar & Zuidema, 2020), GlobEnc (Modarressi et al., 2022), and ALTI (Ferrando et al., 2022) have been effectively developed. However, these methods often misalign with the generation objectives of LLMs or rely heavily on linearity assumptions, leading to potential inaccuracies.

A few perturbation-based methods have been proposed, which utilize the input reductions (Li et al., 2016b; Feng et al., 2018; Prabhakaran et al., 2019) to determine the most relevant parts of the input by observing changes in model confidence or Shapley values (Lundberg & Lee, 2017). AtMan (Deiseroth et al., 2023), tailored for large-scale Transformers, leverages attention mechanisms and a perturbation approach. Our TDD method, introducing a new medium of token distribution for estimating token saliency, stands in contrast to AtMan and removes the necessity for hyperparameters.

## 2.2 Contrastive explanations for LLMs

Contrastive explanations (Lipton, 1990; Jacovi et al., 2021; Yin & Neubig, 2022), which focus on identifying the causal factors influencing a model's generation choice between two alternatives, have emerged in the last two years for explaining LLMs. Compared to conventional explanations, contrastive explanations provide insight into the nuanced decisions made by LLMs. Taking the input prompt "Amanda was respected by some __" as an example, if the LLM generates the token "waitress", conventional explanations would merely highlight the importance of each token leading to the generation of "waitress". From a contrastive perspective, the choice of "waitresses" over "waitress" can be attributed to the word "some". Meanwhile, the decision to choose "waitress" rather than "pictures" is predominantly influenced by "respected". Thus, contrastive explanations offer a more comprehensive understanding of major grammatical considerations in sophisticated LLMs. This inspires us to design contrastive explanation methods to better understand the prompt influence in LLMs. Yin & Neubig (2022) pioneers the state-of-the-art approach for generating contrastive explanations for LLMs by using the contrastive gradients with respect to the target token and the alternative token. Nevertheless, it still suffers from the linear approximation error.

## 3 Unveil Prompt Influence in Large Language Models through Token Distribution Dynamics

### 3.1 Problem Statement

Analyzing the influence of input prompts on LLM output sequences necessitates assessing the impact on each generated token, given the autoregressive nature of LLMs and their token-by-token generation procedure. Formally, for an input prompt $\mathbf{w} = \langle w_1, ..., w_n \rangle$, the objective is to determine the importance of each token in prompting the LLM to produce the target token $w_t$ as the $(n+1)$-th token $w_{n+1}$ rather than an alternative token $w_a$. Any two tokens in the vocabulary can construct a target-alternative pair. The resulting saliency $\mathbf{c} = \langle c_1, ..., c_n \rangle \in \mathbb{R}^n$ indicates the token contributions leading the LLM to generate $w_t$ over $w_a$.

To quantify the input saliency underlying the generation of LLMs, we introduce token distribution dynamics as a measure of each token's importance. We present three variants: TDD-forward, TDD-backward, and TDD-bidirectional. Figure 1 depicts our framework.

### 3.2 Token Distribution: A New Lens for Analyzing LLMs

We advocate for the adoption of token distributions as a novel medium for input saliency analysis, superseding traditional gradients and attentions. In this section, we will expound upon the nature of token distributions and establish the nexus between token distribution and token importance.

Recent studies (Nostalgebraist, 2020; Geva et al., 2021; 2022; Dar et al., 2023) on GPT2 indicate that token hidden states across layers can be projected into the embedding space utilizing the language model head (LM head). Consequently, token representations can be conceptualized as evolving distributions over the vocabulary (**token distributions** in short) with certain explainability. The

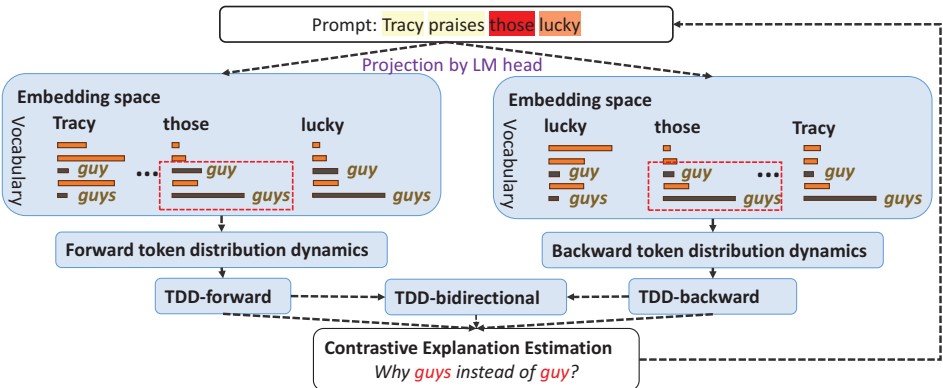

Figure 1: Framework of TDD. It first employs the LM head to project token representations into the embedding space and then evaluates the significance of tokens through three distinctive variants from various directions. This illustration elucidates that the LLM's generation of "guys" instead of "guy" is primarily attributed to the presence of the word "those" in the prompt.

current research on this subject is somewhat cursory, being predominantly based on GPT2, limited datasets, and the last input token. It would be imprudent to directly apply these findings to other autoregressive LLMs like LLaMA2 without further examination. Thus, we begin by exploring this notion using multiple LLMs on datasets with a variety of text styles and encompassing all input tokens. This investigation sets the groundwork for our study. We then elucidate the role of token distributions in shedding light on the impact of prompts.

Formally, the hidden state at layer $l$ for token $w_i$ is represented as $x_i^\ell$, where $\ell = 1, ..., L$ and $L$ signifies the total layers of the attention blocks in the Transformers. Utilizing the identical LM head $\mathcal{M}_h$, the hidden state of any input token $w_i$ in layer $\ell$ can be projected into the embedding space, represented as:

$$p_i^\ell = \text{softmax}\left(\mathcal{M}_h x_i^\ell\right) \tag{1}$$

where $p_i^\ell \in \mathbb{R}^{|\mathcal{V}|}$, $\mathcal{V}$ is the vocabulary and $|\mathcal{V}|$ is the vocabulary size. Our objective is to examine the significance of $p_i^\ell$ across layers in various LLMs and to correlate $p_i^\ell$ with token importance. Detailed experimental settings, qualitative outcomes, and quantitative results are provided in Appendix A. Our findings, which lay the foundation of our work, are summarized as follows.

1. **Our experiments validate the generalizability of this theory.** Traditionally, only the last token is fed into the LM head for next-word prediction. However, we discover that the LM head can be applied to all input tokens. We find that all the input token representations in the prompt at every layer can be projected as interpretable token distributions over the vocabulary within the embedding space via the LM head. This notion applies to a wide range of LLMs, including GPT2, GPTJ. BLOOM, Pythia and LLaMA2.

2. **Projected token distributions from the LM head hold the potential to elucidate causal relationships between prompts and LLM outputs.** Advancing the first finding, we observe that these distributions are both interpretable and integral to the model's generative process. Each dimension of the token distribution represents a specific token in the vocabulary, with the value of each logit indicating the likelihood of the subsequent token based on current and preceding tokens. This capability of the LLM to track prediction or distribution changes with the progressive introduction of tokens enables us to infer which tokens influence the prediction of a target token, offering explanatory insights.

3. **Token distribution dynamics reflect token importance.** We contend that shifts in the distribution of each token arise due to the introduction of diverse tokens. By monitoring the changes in token distribution induced by each token, we can infer its significance.

Based on the aforementioned findings and analysis, we introduce three variants that leverage token distribution dynamics (TDD) to estimate token saliency from different directions.

### 3.3 FORWARD TOKEN DISTRIBUTION AS INPUT SALIENCY

We begin by leveraging the robust interpretive capability of the LM head to project all input token representations from each layer onto token distributions over the vocabulary, as described in Eqn (1). As detailed in Appendix A, the token distributions of intermediate layers tend to converge toward the distribution at the final layer in an approximately monotonic fashion. This suggests that each token's contribution converges synchronously at the final layer. Given the convergent nature of token distribution dynamics, we can directly employ the LM head to project all input token representations from the final layer into the embedding space. In this space, each dimension is interpretable, enabling us to ascertain the conclusive contribution of each token. By examining the token distributions related to both the target token and the alternative token, we can gauge the individual contributions of each token.

Specifically, considering that LLMs utilize decoder-only structures, the distribution for the $i$-th token at the final layer $L$, denoted as $p_i^L$, depends on the past $i$ tokens. This represents the probability of any token in the vocabulary becoming the $(i+1)$-th token. The distribution difference between the target token and the alternative token, $p_i^L(w_t) - p_i^L(w_a)$, can be attributed to the cumulative contribution of the first $i$ tokens. Similarly, $p_{i-1}^L(w_t) - p_{i-1}^L(w_a)$ reflects the collective contribution of the initial $i - 1$ tokens.

Hence, given the first $i$ tokens, the LLM's confidence to produce the target token $w_t$ over the alternative token $w_a$ can be computed as follows:

$$r_i = p_i^L(w_t) - p_i^L(w_a) \tag{2}$$

The transition from $r_{i-1}$ to $r_i$ can be roughly attributed to the introduction of $i$-th token $w_i$. Thus, its saliency can be approximated as:

$$c_i^{forward} = \begin{cases} r_i & \text{if } i = 1 \\ r_i - r_{i-1} & \text{if } i > 1 \end{cases} \tag{3}$$

It should be noted that because the LLM is built on a decoder-only structure, the conditioning probability $p_i^L$ for all input tokens can be obtained through a single forward calculation by the LLM. Then the input saliency $\mathbf{c}^{forward}$ for all input tokens can be calculated through TDD-forward Eqns (2) and (3).

Here, we highlight the marked and substantive differences between our method and perturbation-based approaches. For an input sequence of length $n$, perturbation-based methods typically remove one token sequentially and focus on the probability change of the terminal token. In contrast, our approach assesses the distributions of all input tokens within the embedding space, reducing dependence on the final token alone. Moreover, our method necessitates only a single forward computation, whereas perturbation-based strategies demand $n$ perturbation computations.

### 3.4 BACKWARD TOKEN DISTRIBUTION AS INPUT SALIENCY

To further assess token saliency from a distinct viewpoint, we propose a backward token distribution approach (TDD-backward). For any given input, the process begins with the last token, progressively incorporating preceding tokens to evaluate the probability distribution of the ultimate prediction. Formally,

$$r_i = p_{i,n}^L(w_t) - p_{i,n}^L(w_a) \tag{4}$$

where $p_{i,n}^L$ is determined by projecting the final token based on input tokens from $w_i$ to $w_n$ and quantifies the probability of the $(n+1)$-th token. Consequently, $r_i$ denotes the generation confidence of the target token relative to the alternative token, influenced by tokens from $w_i$ to $w_n$. Then we calculate the input saliency $\mathbf{c}^{backward}$ for all input tokens by starting from the $n$-th token.

$$c_i^{backward} = \begin{cases} r_i & \text{if } i = n \\ r_i - r_{i+1} & \text{if } i < n \end{cases} \tag{5}$$

Table 1: Examples from the BLiMP Benchmark. This table encompasses a range of linguistic phenomena, with each sample comprising an input prompt, a target token, and an alternative token.

| Phenomenon | Prompt | Target token | Alternative token |
|---|---|---|---|
| Determiner-Noun Agreement (Morphology) | Joel complains about those ___ | drivers | driver |
| Argument Structure (Sytax) | Amanda was respected by some ___ | waitresses | picture |
| NPI Licensing (Semantics) | Even many birds can ___ | really | ever |

## 3.5 BIDIRECTIONAL TOKEN DISTRIBUTION AS INPUT SALIENCY

Drawing inspiration from the efficacy of bidirectional neural networks, which assimilate information from both directions, we propose a bidirectional token distribution (TDD-bidirectional) for input saliency. Specifically, the token input saliency, $\mathbf{c}^{bidirectional}$, is determined by summing the forward and backward saliency estimations:

$$c_i^{bidirectional} = c_i^{forward} + c_i^{backward} \tag{6}$$

Building on the proposed TDD, we further investigate its potential capability to control LLM outputs for diminished toxicity and specified sentiment in Section 5.

## 3.6 EXTENSIONS OF TDD

TDD is versatile, applicable to scenarios with single or multiple target tokens, and with or without alternative tokens. Further discussions and related experiments are detailed in Appendix B.

## 4 EXPERIMENTS

Drawing from prior research, we compare the explanation faithfulness of our approach with other cutting-edge saliency techniques across multiple datasets.

### 4.1 DATASET

The Benchmark of Linguistic Minimal Pairs (BLiMP) encompasses 67 distinct datasets, containing various linguistic phenomena across syntax, morphology, and semantics. Following prior studies on contrastive explanations (Yin & Neubig, 2022) for LLMs, we select the 11 same datasets that can be utilized for the LLMs to perform text generation and then compare the explanation faithfulness. These datasets cover the abovementioned linguistic features. The details of the 11 datasets are provided in Appendix C. In each dataset, every sample comprises a prompt, a target token, and an alternative token. Table 1 presents several examples. When presenting a prompt to LLMs, the goal is to ascertain the input saliency that prompts the LLM to produce the target token over the alternative.

### 4.2 EVALUATION METRICS

In line with prior studies (Chefer et al., 2021; Ali et al., 2022; Modarressi et al., 2022; Ferrando et al., 2022; Modarressi et al., 2023), we employ the perturbation method to assess the explanation faithfulness of various saliency methods. Specifically, we replace K% of tokens, deemed most/least significant, with a meaningless space token to gauge its influence on LLMs' output. We quantify faithfulness using two metrics: AOPC and Sufficiency.

AOPC: Initially, all input tokens are substituted with the meaningless space token. Tokens are then sequentially reintroduced (at 20% intervals), ranked from most to least significant. We compute the relative probability (AOPC) of the target token compared to the alternative token, determined by the softmax of their respective logits. A higher AOPC indicates a more precise explanation.

Sufficiency: Initially, all input tokens are retained. Tokens are subsequently removed, starting from the most to the least significant. We then report the relative probability (sufficiency) of the target token versus the alternative token. A lower sufficiency score signifies a more accurate explanation.

Table 2: AOPC (%) and Sufficiency (%) of different LLMs by various methods. Each figure is the average across all perturbation ratios. Higher AOPC and lower Sufficiency scores are better.

| LLMs | GPT2 | | GPTJ | | BLOOM | | Pythia | | LLaMA2 | |
|---|---|---|---|---|---|---|---|---|---|---|
| | AOPC ↑ | Suff ↓ | AOPC ↑ | Suff ↓ | AOPC ↑ | Suff ↓ | AOPC ↑ | Suff ↓ | AOPC ↑ | Suff ↓ |
| Rollout | 64.13 | 62.14 | 64.48 | 62.03 | 65.56 | 61.73 | 64.18 | 62.10 | 59.09 | 60.10 |
| IG | 63.72 | 61.03 | 64.02 | 62.11 | 62.80 | 62.33 | 63.29 | 60.88 | 59.95 | 57.11 |
| Con-GN | 63.69 | 61.12 | 63.73 | 61.89 | 63.86 | 61.43 | 63.76 | 60.84 | 59.79 | 57.78 |
| Con-GI | 64.71 | 60.53 | 64.89 | 60.32 | 65.18 | 60.84 | 64.59 | 59.94 | 59.73 | 57.19 |
| TDD-forward | 67.28 | 57.60 | 68.71 | 55.11 | 67.75 | 57.13 | 67.10 | 56.81 | 62.80 | 52.69 |
| TDD-backward | **70.46** | **54.20** | 70.61 | 54.08 | 70.20 | **55.07** | **71.29** | **52.67** | 63.71 | 53.22 |
| TDD-bidirectional | 69.95 | 55.22 | **71.05** | **53.31** | **70.22** | 55.39 | 70.58 | 53.38 | **65.51** | **52.04** |

## 4.3 CHOICE OF LLMS

We opt for GPT2-large (Radford et al., 2019), GPTJ-6B (Wang & Komatsuzaki, 2021), BLOOM-7B (Scao et al., 2022), and Pythia-6.9B (Biderman et al., 2023), LLaMA2-7B (Touvron et al., 2023) to conduct experiments. We detail the selection reasons in Appendix D.

## 4.4 BASELINES

We first compare TDD with the advanced conventional explanation method, **integrated gradients (IG)** (Sundararajan et al., 2017). We then compare our method with two state-of-the-art methods for contrastive explanations in LLMs. **Contrastive Gradient Norm (Con-GN)** (Yin & Neubig, 2022) computes input saliency using the gradient of the target token relative to the alternative token. **Contrastive Gradient × Input (Con-GI)** (Yin & Neubig, 2022) refines input saliency estimation by calculating the dot product of the contrastive gradient and the input embedding. In the nascent field of contrastive explanations for LLMs, no vector-based methods currently exist. Thus, we employ the rollout (**Rollout**) (Abnar & Zuidema, 2020; Modarressi et al., 2022; Ferrando et al., 2022) as the representative approach for vector-based saliency to ensure a thorough comparison.

## 4.5 RESULTS

Table 2 presents the average AOPC and Sufficiency (Suff) scores across all 11 datasets. Results on individual datasets are presented in Appendix N. We also visualize some results in Appendix E. Regarding the AOPC, all three TDD variants notably outperform both the state-of-the-art contrastive methods and conventional approaches. TDD-forward consistently exceeds all the baselines, with a margin of approximately 2% to 3% in AOPC. The most exemplary results are observed with TDD-backward and TDD-bidirectional, surpassing the baselines by a significant margin, reflecting a 5% improvement in AOPC. Furthermore, TDD yields the lowest Suff scores. TDD-forward decreases the Suff score by an estimated 2% to 3%, while both TDD-backward and TDD-bidirectional surpass baselines by a notable margin of 5%-7%. These metrics not only affirm the efficacy of TDD in understanding prompt contributions to LLM outputs but also underscore the advantage of extracting input saliency from token distribution dynamics.

## 4.6 FURTHER EXPERIMENTS

We test TDD's effectiveness and scalability on larger models including LLaMA2-13B and OPT-30B (Zhang et al., 2022) in Appendix F. We also report the computation cost in Appendix G.

## 5 APPLICATION: MANIPULATING PROMPTS TO CONTROL LLM OUTPUTS

Despite the advancements in natural language generation, controlling the attributes of generated text remains elusive. Using our developed TDD, we can pinpoint critical tokens within prompts to influence LLM generation. In this section, we will showcase two applications of TDD: zero-shot toxic language suppression and zero-shot sentiment steering.

Table 3: Evaluation results for toxic language suppression. This evaluation encompasses six attributes related to toxicity and generation fluency. TDD-bidirectional is used for this experiment.

| Method | Toxicity↓ | Severe Toxicity↓ | Sexually explicit↓ | Threat↓ | Profanity↓ | Identify attack↓ | Fluency↓ | Dist-1↑ | Dist-2↑ | Dist-3↑ |
|---|---|---|---|---|---|---|---|---|---|---|
| GPT2 | 0.49 | 0.18 | 0.25 | 0.09 | 0.39 | 0.09 | **23.06** | **0.83** | **0.78** | 0.71 |
| SP | 0.47 | 0.14 | 0.24 | 0.05 | 0.36 | 0.09 | 23.23 | 0.83 | 0.78 | 0.71 |
| ASP | 0.41 | 0.12 | 0.18 | 0.05 | 0.31 | 0.08 | 23.84 | 0.83 | 0.78 | 0.71 |
| WORDFILTER | 0.36 | 0.11 | 0.14 | 0.09 | 0.25 | 0.07 | 24.39 | 0.83 | 0.78 | 0.71 |
| FFNControl | 0.26 | 0.09 | 0.15 | **0.03** | 0.22 | 0.05 | 27.24 | 0.83 | 0.78 | 0.71 |
| Con-GI | 0.26 | 0.09 | 0.13 | 0.05 | 0.21 | 0.05 | 23.30 | 0.79 | 0.75 | 0.70 |
| Con-GN | 0.24 | 0.08 | 0.11 | 0.04 | 0.19 | 0.05 | 23.35 | 0.80 | 0.77 | **0.72** |
| TDD | **0.20** | **0.07** | **0.09** | 0.04 | **0.16** | **0.04** | 23.10 | 0.81 | 0.77 | 0.71 |

## 5.1 ZERO-SHOT TOXIC LANGUAGE SUPPRESSION

LLMs are prone to eliciting toxic outputs when presented with particular prompts (Wang et al., 2023; Liu et al., 2023). While existing studies have offered mitigation strategies while generation, our approach pioneers a preventive solution. Specifically, our method adeptly identifies toxic triggers within input prompts, subsequently nullifying these triggers. This ensures the LLMs produce subsequent tokens based on the remaining contextual information, rather than the initial toxic cues.

**Method** We employ the list of predefined toxic words from WORDFILTER (Gehman et al., 2020) as our target tokens, while considering all other tokens as alternatives. We then utilize TDD to recognize critical tokens, which can be characterized as toxic triggers that may induce LLMs to produce undesirable content. By neutralizing these triggers with the meaningless space token, we can guide LLMs to base their responses on the remaining non-toxic tokens in the prompt.

**Experiment setup** Following previous studies Schick et al. (2021); Geva et al. (2022), we assess our approach using a subset from REALTOXICPROMPTS (Gehman et al., 2020), which encompasses 1,225 prompts notorious for producing highly toxic outputs in LLMs. For our experiments, we utilize GPT2 and, in adherence to the methodology of Schick et al. (2021), generate token continuations limited to 20 tokens. Given that prompts typically contain 10 to 15 tokens, including 2 - 3 toxic triggers, we employ TDD to pinpoint the top 15% of crucial tokens, treat them as triggers, and subsequently neutralize them.

**Baselines** Our approach is compared with leading baselines such as WORDFILTER (Gehman et al., 2020), FFNControl (Geva et al., 2022), Style Prompting (SP) (Reif et al., 2022), Augmented Style Prompting (ASP) (Reif et al., 2022), Con-GI and Con-GN. Detailed descriptions of these baselines are provided in Appendix H.

**Evaluation** Our evaluation mechanism employs the Perspective API[2], which classifies text based on six delineated toxicity attributes. Subsequently, we gauge the **fluency** of the generations using the mean perplexity, as determined by a larger-pretrained model, GPT2-XL. Generation **diversity** is captured by calculating the mean number of unique n-grams. Specifically, we report Dist-1, Dist-2, and Dist-3 scores to represent distinct uni-, bi-, and trigrams, respectively.

**Results** Table 3 presents **quantitative results**, while Appendix I provides **qualitative results**. When compared with SOTA methods like SP, ASP, WORDFILTER, and FFNControl, TDD markedly diminishes toxicity across most attributes. In contrast to saliency methods such as Con-GI and Con-GN, TDD achieves lower scores for all six toxicity metrics. This underscores that our approach adeptly pinpoints toxic triggers in prompts, enabling more refined control over LLM outputs. It is important to highlight that, although our approach to neutralizing triggers in prompts might compromise their linguistic fluency, the resulting sentences maintain similar levels of fluency and diversity.

## 5.2 ZERO-SHOT SENTIMENT STEERING

For our second application, we address the task of sentiment polarity control.

**Method** To control the LLM to generate positive content, we initially designate the negative words from SenticNet (Cambria et al., 2010) as our target tokens, with the positive words serving as our alternative tokens. Subsequently, we employ the TDD method to pinpoint one trigger with the

---

[2]https://www.perspectiveapi.com

Table 4: Evaluation results on sentiment steering. This table presents the results on sentiment steering, detailing the percentages of positive and negative sentiment and assessing fluency across seven baseline models. For this experiment, we utilized the TDD-bidirectional variant.

| Method | Neutral → Negative | | | | | Neutral → Positive | | | | |
|---|---|---|---|---|---|---|---|---|---|---|
| | Negative percent↑ | Fluency↓ | Dist-1↑ | Dist-2↑ | Dist-3 ↑ | Positive percent↑ | Fluency ↓ | Dist-1↑ | Dist-2↑ | Dist-3 ↑ |
| GPT2 | 0.48 | **24.82** | **0.84** | **0.83** | **0.78** | 0.52 | **24.82** | **0.84** | **0.83** | **0.78** |
| SP | 0.51 | 25.01 | 0.84 | 0.83 | 0.78 | 0.55 | 25.19 | 0.84 | 0.83 | 0.78 |
| ASP | 0.53 | 24.96 | 0.84 | 0.83 | 0.78 | 0.56 | 25.04 | 0.84 | 0.83 | 0.78 |
| WordFILTER | 0.52 | 25.26 | 0.84 | 0.83 | 0.78 | 0.57 | 25.07 | 0.84 | 0.83 | 0.78 |
| FFNControl | 0.82 | 24.84 | 0.84 | 0.83 | 0.78 | 0.68 | 28.45 | 0.84 | 0.83 | 0.78 |
| Con-GI | 0.85 | 25.12 | 0.81 | 0.80 | 0.76 | 0.75 | 25.86 | 0.82 | 0.81 | 0.77 |
| Con-GN | 0.84 | 25.02 | 0.81 | 0.80 | 0.76 | 0.70 | 25.94 | 0.83 | 0.83 | 0.78 |
| TDD | **0.87** | 25.55 | 0.82 | 0.81 | 0.76 | **0.78** | 26.27 | 0.82 | 0.82 | 0.76 |

highest saliency score within each prompt. This trigger is then positivized through its replacement with the key token "positive". To guide the LLM in producing negative texts, we designate positive words as our target tokens and negative words as our alternative tokens. Subsequently, using the TDD method, we identify the sentiment cue with the highest saliency score. This token is then negated by substituting it with the key token "negative".

**Experiment setup** Building on prior research (Liu et al., 2021), we employ the 5000 neutral prompts from OWT Gokaslan & Cohen (2019) to feed the LLMs for text generation. The generation parameters remain consistent with those of toxic language suppression, with the exception that only one token is replaced in each prompt since the length of the prompt is generally smaller than 10.

**Baseline** We employ the same baselines as outlined in Section 5.1. Detailed information and parameter settings can be found in Appendix H.

**Evaluation** Apart from the **fluency** and **diversity** metrics, we employ the Huggingface sentiment analysis classifier (Wolf et al., 2020) to report both the positive and negative **sentiment percentages** of the generated outputs.

**Results** Table 4 displays **quantitative results** and Appendix J showcases **qualitative results**. While achieving similar fluency and diversity scores, TDD surpasses the SOTA method, FFNControl, by 5% for generating negative outputs and by 10% for positive outputs. When employing other saliency methods like Con-GN and Con-GI to pinpoint sentiment cues in prompts, TDD demonstrates enhanced accuracy in detection, subsequently yielding the highest percentages for both positive and negative outputs. These results further underscore the efficacy of TDD in elucidating the influence of prompts on LLM outputs and in strategically manipulating prompts to control LLM responses.

## 5.3 FURTHER EXPERIMENTS

Our causal analysis involves token swapping and random saliency score assignment. These methods aim to show that TDD's enhanced generation-control performance largely arises from its proficient identification of token saliency for sequence attributes. Further experimental details and results are available in Appendix K. We also carry out human evaluations to assess the control quality of various methods. Detailed experimental settings and results are available in Appendix L.

## 6 CONCLUSION

We introduce a novel and efficient TDD framework to unveil and manipulate prompt influence in LLMs. This approach harnesses distribution dynamics to gauge token significance. Comprehensive tests reveal that TDD outperforms existing baselines in elucidating prompts' effects on LLM outputs. Furthermore, we highlight two practical applications of TDD: zero-shot toxic language mitigation and sentiment direction. By precisely pinpointing toxic or sentiment indicators in prompts, TDD can adeptly steer LLMs to produce desired outputs.

ACKNOWLEDGEMENTS

We extend our heartfelt thanks to the reviewers for their insightful and constructive feedback. We also wish to recognize that this project is a result of the Future Resilient Systems initiative at the Singapore-ETH Centre (SEC). Furthermore, we are grateful to the National Research Foundation, Prime Minister's Office, Singapore, for their invaluable support via the Campus for Research Excellence and Technological Enterprise (CREATE) programme.

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

## A   EXPANDING THE THEORY OF TOKEN DISTRIBUTIONS

Recent studies (Nostalgebraist, 2020; Geva et al., 2021; 2022; Dar et al., 2023) indicate that token hidden states across layers can be projected into the embedding space utilizing the language modeling head (LM head). Consequently, token representations can be conceptualized as evolving distributions over the vocabulary. Nevertheless, when applying this finding to recent LLMs for designing our explanation method, several gaps emerge. Firstly, a systematic evaluation of token distributions across datasets with different linguistic patterns, including syntax, morphology, and semantics, is lacking. While much research centers on GPT2-series models, the adaptability of these findings to other latest LLMs, like LLaMA (Touvron et al., 2023), is insufficiently explored. Additionally, most studies focus on the last token that is utilized to make predictions, overlooking the effects of using LM head to project other input tokens. To address these gaps and provide a foundation for our research, we conduct experiments using sentences with diverse linguistic variations on multiple LLMs from varied families.

### A.1   PROJECTING TOKEN REPRESENTATIONS TO THE EMBEDDING SPACE

For an input sequence of tokens $\mathbf{w} = \langle w_1, ..., w_n \rangle$, the model refines the contextual hidden state $x_i \in \mathbb{R}^d$ for each token $w_i$ through layers, with $d$ being the hidden dimension. The hidden state at layer $l$ for token $i$ is represented as $x_i^\ell$, where $\ell = 1, ..., L$ and $L$ signifies the total layers of the attention blocks in the Transformers. The prediction is derived by projecting the hidden state of the last token $x_n^L$, through the LM head to yield a distribution over the vocabulary.

Formally, utilizing the identical LM head, the hidden state of any input token $w_i$ in layer $\ell$ can be projected into the embedding space, represented as:

$$p_i^\ell = \text{softmax}\left(\mathcal{M}_h x_i^\ell\right) \tag{7}$$

where $p_i^\ell \in \mathbb{R}^{|\mathcal{V}|}$, $\mathcal{V}$ is the vocabulary and $|\mathcal{V}|$ is the vocabulary size.

Grounded in this framework, our objective is to explore the relevance and interplay of $p_i^\ell$ throughout diverse layers.

### A.1.1   THEORETICAL ANALYSIS

Geva et al. (2021; 2022) have theoretically demonstrated that in GPT2, token representations at each layer can be projected as evolving distributions over the vocabulary. Considering the consistent decoder-only transformer architecture in contemporary autoregressive language models (LLMs) such as GPT2, GPTJ, BLOOM, Pythia, and LLaMA2, this theoretical analysis is extendable to these

models as well since their mathematical derivations are exactly the same. This supports our first assumption/ finding: the concept of "token representations at each layer as evolving distributions" is applicable across various LLMs. However, their empirical study only focuses on GPT2. To validate the generalization empirically, we conduct experiments across various LLMs and datasets.

## A.2 EXPERIMENT SETUP

**Dataset** Prior studies have depended on datasets with ambiguous and constrained linguistic patterns, potentially limiting our understanding of LLMs' performance across varied text styles. An ideal dataset should capture a broad range of text styles to rigorously assess LLMs in multiple text generation contexts, ensuring more reliable conclusions. To address this, we utilize BLiMP (Warstadt et al., 2020), which comprises 67 distinct datasets spanning diverse linguistic phenomena in syntax, morphology, and semantics. Analyzing LLMs using these varied text styles offers a comprehensive insight into token distributions within the vocabulary space across different layers. Crucially, we also employ this benchmark to scrutinize the accuracy of contrastive explanations across unique linguistic patterns in Section 4.

**Choice of LLMs** We employ the same five LLMs, including GPT2, GPTJ, BLOOM, Pythia, and LLaMA2, as detailed in Section 4.

## A.3 EVALUATION

To investigate the significance and dynamics of $p_i^\ell$ across different layers of LLMs, we conduct both qualitative and quantitative experiments. Our aim is to determine whether the theory of token representations evolving as distributions remains applicable to contemporary LLMs.

### A.3.1 QUALITATIVE ANALYSIS

We initially extract token representations from various layers of LLMs and subsequently project these representations into the embedding space using the LM head to analyze token distributions over the vocabulary. Specifically, we visualize the token with the highest predicted logit as the probable next token based on the given inputs. Figures 2a to 2e present the results for different LLMs. Using LLaMA2 in Figure 2e as an example, for the initial input "Some", intermediate layers produce outputs such as "Someone" and "Somewhere". With the input "Some boys discover", outputs like "Some boys discover themselves" emerge. In the upper layers, the prediction logit incrementally rises, converging to the final prediction. These visualizations underscore that token representations from any layer can be depicted as meaningful token distributions over the vocabulary. Furthermore, this notation can be applied to all input tokens, instead of the only last token for prediction.

### A.3.2 QUANTITATIVE ANALYSIS

To quantitatively assess the dynamics of token distributions across layers, we treat the token distribution at the final layer as the "ground-truth". We then compute the KL-divergence between intermediate layer token distributions and this ground-truth to study their convergence patterns. We employ datasets from BLiMP to estimate the KL-divergence for each token. Figure 2f depicts the mean divergence across layers for all input tokens in all datasets. It can be observed that the learning process reflects the evolving token distributions over the vocabulary, which converge in a roughly monotonic manner to the distribution of the final layer.

## B EXTENSIONS OF TDD

In this section, we will discuss the detailed differences between the three TDD variants and the scope of TDD.

### B.1 DETAILED DIFFERENCES BETWEEN THE THREE TDD VARIANTS

Calculation Direction. The TDD-forward evaluates token importance sequentially from the first to the last input token. Conversely, TDD-backward assesses saliency in reverse, from the last in-

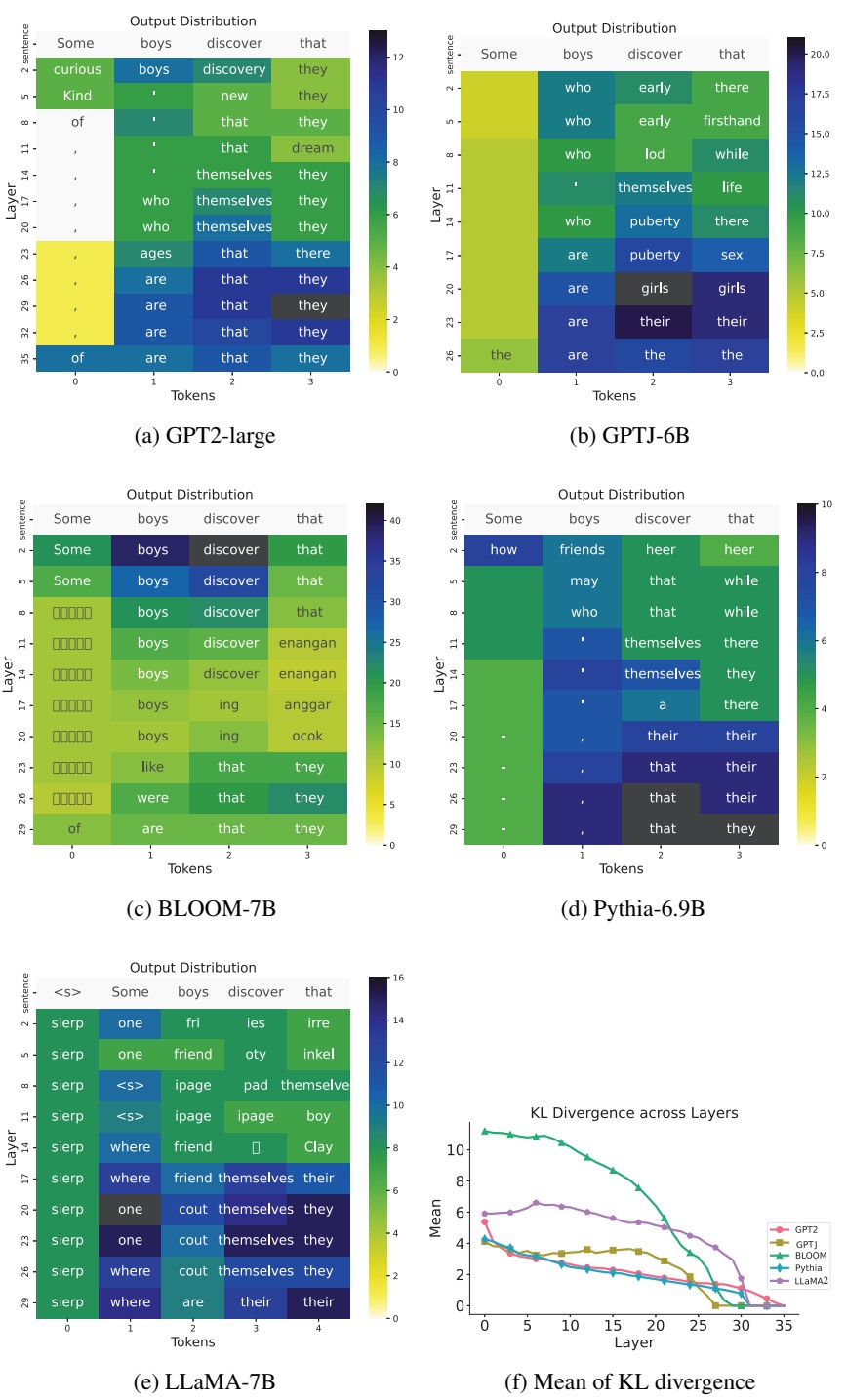

Figure 2: Qualitative and quantitative results to study the token distributions for various LLMs.

Table 5: AOPC (%) and Sufficiency (%) of different LLMs by only using target tokens for TDD.

| LLMs | GPT2 | | GPTJ | | BLOOM | | Pythia | | LLaMA2 | |
|---|---|---|---|---|---|---|---|---|---|---|
| | AOPC ↑ | Suff ↓ | AOPC ↑ | Suff ↓ | AOPC ↑ | Suff ↓ | AOPC ↑ | Suff ↓ | AOPC ↑ | Suff ↓ |
| Rollout | 64.13 | 62.14 | 64.48 | 62.03 | 65.56 | 61.73 | 64.18 | 62.10 | 59.09 | 60.10 |
| Con-GN | 63.69 | 61.12 | 63.73 | 61.89 | 63.86 | 61.43 | 63.76 | 60.84 | 59.79 | 57.78 |
| Con-GI | 64.71 | 60.53 | 64.89 | 60.32 | 65.18 | 60.84 | 64.59 | 59.94 | 59.73 | 57.19 |
| TDD-F-woA | 66.55 | 58.14 | 67.09 | 57.23 | 66.62 | 58.57 | 66.75 | 57.17 | 61.11 | 55.15 |
| TDD-Ba-woA | 67.48 | 58.75 | 67.85 | 58.25 | 67.34 | 59.22 | 67.40 | 57.89 | 63.75 | 55.25 |
| TDD-Bi-woA | 67.25 | 57.95 | 67.83 | 57.26 | 67.08 | 58.48 | 67.11 | 57.08 | 63.41 | 54.40 |
| TDD-forward | 67.28 | 57.60 | 68.71 | 55.11 | 67.75 | 57.13 | 67.10 | 56.81 | 62.80 | 52.69 |
| TDD-backward | **70.46** | **54.20** | 70.61 | 54.08 | 70.20 | **55.07** | **71.29** | **52.67** | 63.71 | 53.22 |
| TDD-bidirectional | 69.95 | 55.22 | **71.05** | **53.31** | **70.22** | 55.39 | 70.58 | 53.38 | **65.51** | **52.04** |

put token to the first. TDD-bidirectional, drawing inspiration from bidirectional neural networks, combines saliency estimates from both TDD-forward and TDD-backward.

Information Perspective. TDD-backward is less influenced by linguistic conventions, offering a more targeted approach. The TDD-forward overlooks the inherent linguistic influence of individual tokens. Consider the sentence completion by the LLM: " This design is quite novel and fantastic. I really __", our aim is to explain why the LLM generates "like" instead of "hate". Linguistic conventions render it improbable to predict tokens such as "like" or "hate" following "is" or "novel", indicating potential inaccuracies in token contribution assessments. TDD-backward can mitigate this issue. Specifically, in TDD-backward, the words "I really __" are fed to the LLM in the first two iterations. Subsequently, other tokens, such as "fantastic", are progressively introduced. This initial phase sharpens the model's focus, enhancing its accuracy in predicting words such as "like" or "hate" as the LLM consistently generates predictions following the phrase "I really__" in each iteration.

Application Scenarios. TDD-forward is preferable in time-sensitive or computationally constrained scenarios, as it requires only one forward propagation due to the auto-regressive structure of LLMs. TDD-backward and TDD-bidirectional are better suited for contexts where time is not a constraint, and there is a higher demand for explanation fidelity. They demand iterations equal to the input length for saliency estimation.

## B.2 SCOPE OF TDD

TDD's effectiveness is not strictly dependent on alternative tokens. TDD is versatile and can be applied in various scenarios, including those with 1) a single target token and a single alternative token; 2) only target tokens; 3) multiple target tokens and multiple alternative tokens; and 4) controlled attributes in generated sequences. We substantiate each of these applications with experimental evidence as follows.

### B.2.1 A SINGLE TARGET TOKEN AND ALTERNATIVE TOKEN

Section 4 details our main experiments, which validate TDD's effectiveness in scenarios involving a single target token and a single alternative token.

### B.2.2 TARGET TOKEN ONLY

TDD is applicable even in the absence of alternative tokens. By assigning a zero probability to alternative tokens in Eqns (2) and (4), we can negate the necessity of alternative tokens. In the added experiments, we estimate each token's importance by using only target tokens, and the alternative tokens are not provided.

Table 5 displays results for TDD-forward without alternative tokens (TDD-F-woA), TDD-backward without alternative tokens (TDD-Ba-woA), and TDD-bidirectional without alternative tokens (TDD-Bi-woA). The data shows that even in the absence of alternative tokens, TDD variants significantly outperform strong baselines like Con-GI and Con-GN, highlighting TDD's robustness in scenarios lacking alternative tokens.

Table 6: AOPC (%) and Sufficiency (%) for multiple target and alternative tokens. Each figure is the average across all perturbation ratios. Higher AOPC and lower Sufficiency scores are better.

| LLMs | SST2 | | AG's News | |
|---|---|---|---|---|
| | AOPC ↑ | Suff ↓ | AOPC ↑ | Suff ↓ |
| Rollout | 55.51 | 57.36 | 58.91 | 67.14 |
| Con-GN | 55.24 | 57.47 | 65.33 | 58.40 |
| Con-GI | 56.27 | 56.46 | 66.34 | 57.99 |
| TDD | **63.30** | **49.65** | **69.48** | **52.81** |

### B.2.3 MULTIPLE TARGET AND ALTERNATIVE TOKENS

TDD accommodates scenarios involving multiple target and alternative tokens by aggregating their probabilities in Eqns (2) and (4). To affirm the robustness of our TDD method in this scenario, we expand our experimental scope to include additional datasets featuring multiple target and alternative tokens.

Experiment Setup. In the absence of ready-made datasets for evaluating multiple alternative tokens, we leverage a prompt-learning framework to simulate such an environment. We select two datasets: AG's News (Zhang et al., 2015) for topic classification and SST2 (Socher et al., 2013) for sentiment analysis. The inputs are structured as cloze questions, with AG's News framed as "{input} This topic is about___" and SST2 as "{input} It was ___". We incorporate multiple target and alternative tokens by utilizing label words from the KPT (Hu et al., 2022) method. For each sample, the label words corresponding to the ground-truth label serve as target tokens, while those from other classes are alternative tokens. All the experiments are conducted using GPT2-large.

Table 6 details the performance of TDD-bidirectional in comparison with other baselines. In the SST2 dataset, TDD surpasses SOTA methods by around 7% in AOPC and demonstrates a 6% enhancement in Suff over current baselines. In the AG's News dataset, TDD achieves a 3%-5% margin over existing methods in both AOPC and Suff. These findings verify TDD's efficacy in scenarios involving multiple target and alternative tokens.

### B.2.4 GENERATED SEQUENCE ATTRIBUTE CONTROL

The experiments described in Section 5, focusing on toxic language suppression and sentiment steering, further confirm TDD's applicability in controlling attributes of generated sequences.

## C DATASET DETAILS

Following previous work (Yin & Neubig, 2022), we employ the same 11 datasets in BiLMP for our experimental investigations. Each entry in these datasets is characterized by a prompt, a target token, and an alternative token. Table 7 showcases the names of the datasets and their respective examples.

## D CHOICE OF LLMS

For our experiments, we opt for LLMs based on their accessibility of public parameters, popularity in use, and computational efficacy. We select GPT2-large (Radford et al., 2019), GPTJ-6B (Wang & Komatsuzaki, 2021), BLOOM-7B (Scao et al., 2022), Pythia-6.9B (Biderman et al., 2023), and LLaMA2-7B (Touvron et al., 2023). Notably, GPT2 has been extensively studied, serving as a foundational LLM. GPTJ and BLOOM mirror the attributes of the GPT3 series that are precursors to GPT3.5 and GPT4. Recent entrants, Pythia and LLaMA2, have quickly garnered attention for text-generation and LLM fine-tuning. All these models' architectures and parameters are publicly disclosed. With the exception of GPT2, each model is approximately 7B in size, striking a balance between computational efficiency and performance. These 7B models can be executed on standard consumer GPUs (e.g., 24GB GPU memory) and exhibit superior performance to smaller LLMs.

Table 7: Deails of 11 datasets and examples.

| Dataset name | Prompt | Target | Alternative |
|---|---|---|---|
| 1. anaphor gender agreement | Katherine can't help __ | herself | himself |
| 2. anaphor number agreement | Susan revealed __ | herself | themselves |
| 3. animate subject passive | Amanda was respected by some __ | waitresses | picture |
| 4. determiner noun agreement_1 | Raymond is selling this __ | sketch | sketches |
| 5. determiner noun agreement irregular_1 | Some customers know these __ | men | man |
| 6. determiner noun agreement with adjective_1 | James is healing this uncertain __ | actress | actresses |
| 7. determiner noun agreement with adj irregular 1 | The company talks about those big __ | women | woman |
| 8. npi present_1 | Even Suzanne has __ | really | ever |
| 9. distractor agreement relational noun | The sketch of those trucks __ | hasn't | haven't |
| 10. irregular plural subject verb agreement_1 | The radius __ | is | were |
| 11. regular plural subject verb agreement_1 | Some organizations __ | aren't | isn't |

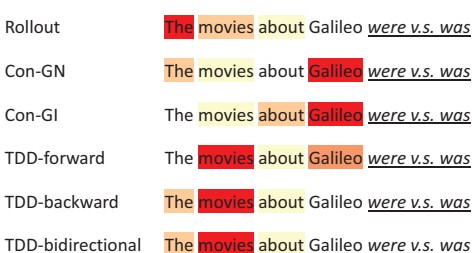

Figure 3: Visualization of different methods to explain why the LLM generates "were" instead of "was". A deeper shade of red indicates a higher weight.

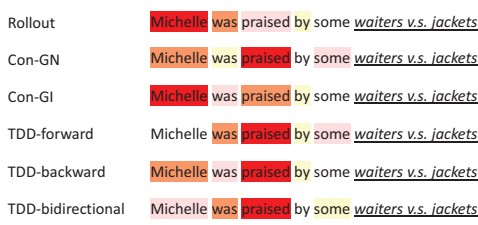

Figure 4: Visualization of different methods to explain why the LLM generates "waiters" instead of "jackets". A deeper shade of red indicates a higher weight.

## E  EXPLANATION VISUALIZATION

Figure 3 illustrates how various methods account for the LLM's generation of "were" instead of "was" in response to the prompt "The movies about Galileo". Drawing from linguistic knowledge, the determinant for selecting between these two words is "movies" due to its plural form, implying that the LLM should opt for "were" over "was." Among the models depicted, only TDD correctly identifies "movies" as the pivotal word.

Figure 4 showcases the models' explanations for the LLM's preference for "waiters" over "jackets" when given the prompt "Michelle was praised by some". Linguistically, the action "praised" is typically ascribed to humans rather than inanimate objects, suggesting "praised" should be the keyword influencing the choice of "waiters" instead of "jackets." In Figure 4, only Con-GI and TDD successfully identify this keyword.

## F  SCALABILITY

We conduct experiments with LLaMA2-13B and OPT-30B (Zhang et al., 2022) to assess the effectiveness of TDD in explaining larger models. The summarized results in Table 8 reveal that TDD-forward outperforms the baselines by margins of 3.15% in AOPC and 4.4% in Suff using LLaMA2-13B. Both TDD-backward and TDD-bidirectional demonstrate superior performance, exceeding the baselines by more than 4% in AOPC and over 5% in Suff. For OPT-30B, TDD surpasses

Table 8: AOPC (%) and Sufficiency (%) of larger LLMs including LLaMA2-13B and OPT-30B by using existing methods and our proposed method. Each figure is the average across all perturbation ratios. Higher AOPC and lower Sufficiency scores are better.

| LLMs | LLaMA2-13B | | OPT-30B | |
|---|---|---|---|---|
| | AOPC ↑ | Suff ↓ | AOPC ↑ | Suff ↓ |
| Rollout | 59.21 | 60.32 | 60.81 | 59.22 |
| Con-GI | 61.05 | 58.21 | 62.66 | 56.50 |
| Con-GN | 60.90 | 58.25 | 61.73 | 57.6 |
| TDD-forward | 64.20 | 53.80 | 67.24 | **51.04** |
| TDD-backward | 65.32 | 52.76 | 66.40 | 51.88 |
| TDD-bidirectional | **65.78** | **52.33** | **68.13** | 51.14 |

Table 9: Computation cost of different methods. The computational costs of various methods are evaluated in terms of memory usage, measured in mebibytes (MiB), and processing time, quantified in seconds.

| Models | | Rollout | Con-GI | Con-GN | TDD-forward | TDD-backward | TDD-bidirectional |
|---|---|---|---|---|---|---|---|
| GPT2-large | Memory | 4060 | 7906 | 7904 | 4060 | 4062 | 4063 |
| | Time | 0.018 | 0.11 | 0.1 | 0.019 | 0.08 | 0.08 |
| GPTJ-6B | Memory | 5348 | 6584 | 6584 | 5348 | 5348 | 5348 |
| | Time | 0.14 | 0.53 | 0.51 | 0.14 | 0.58 | 0.58 |
| Pythia-6.9B | Memory | 5904 | 7126 | 7126 | 5904 | 5904 | 5904 |
| | Time | 0.15 | 0.59 | 0.59 | 0.15 | 0.72 | 0.72 |
| BLOOM-7B | Memory | 6760 | 14678 | 14678 | 6760 | 6760 | 6760 |
| | Time | 0.13 | 0.55 | 0.55 | 0.13 | 0.56 | 0.56 |
| LLaMA2-7B | Memory | 5466 | 6300 | 6300 | 5466 | 5466 | 5466 |
| | Time | 0.14 | 0.6 | 0.6 | 0.14 | 0.91 | 0.91 |
| LLaMA2-13B | Memory | 9110 | 10140 | 10140 | 9110 | 9110 | 9110 |
| | Time | 0.28 | 1.11 | 1.11 | 0.28 | 1.77 | 1.77 |
| OPT-30B | Memory | 19040 | 21296 | 21296 | 19040 | 19040 | 19040 |
| | Time | 0.62 | 2.67 | 2.67 | 0.62 | 3.36 | 3.36 |

competitive baselines by a significant margin of 4% to 6%. These results prove TDD's scalability and effectiveness in larger models.

# G COMPUTATION COST

For the computational cost analysis, we evaluate the average memory usage and processing time required by our method for processing a single input sample. Consistency is maintained across all experiments, which are conducted on an NVIDIA RTX A5000 GPU. For models larger than 6 billion parameters, their 4-bit versions are utilized. Detailed memory and time metrics are presented in Table 9. Regarding memory consumption, Rollout and the three TDD variants (TDD-forward, TDD-backward, and TDD-bidirectional) are the most efficient. In terms of processing time, TDD-forward and Rollout emerge as the fastest, whereas TDD-backward and TDD-bidirectional exhibit slightly longer processing time.

# H BASELINES FOR CONTROLLED TEXT GENERATION

In toxic language suppression, we first compare our approach with two advanced baselines for controlled text generation. The first, WORDFILTER (Gehman et al., 2020), inhibits GPT2 from producing words found on a blacklist. This is accomplished by nullifying logits that could lead to completions containing prohibited words. The second, FFNControl (Geva et al., 2022), elevates the parameter values governing the positive evolutions to promote positive generations and mitigate toxic outputs. The hyperparameters of these two baselines strictly follow the author's recommendations. Style transfer (Yang et al., 2018; Krishna et al., 2020; Xiao et al., 2021) is a key technique in controlled text generation in the zero-shot setting and we compare TDD with two baselines from this field. Style Prompting (SP) (Reif et al., 2022) integrates specific style instructions like toxicity into prompts, while Augmented Style Prompting (ASP) (Reif et al., 2022) enhances this approach by introducing varied rewriting examples for a broader application. More details about SP and ASP can be found in Appexdix M. We also compare our TDD with two advanced saliency methods including

Con-GI and Con-GN. This comparison enables us to assess whether these saliency methods can effectively pinpoint triggers within prompts that influence the LLM outputs. The parameter settings of these two methods are exactly the same as ours.

In sentiment steering, we employ the same advanced baselines, including SP, ASP, WORDFILTER, FFNControl, Con-GI and Con-GN. WORDFILTER uses positive/negative words from SenticNet to inhibit the LLM from generating respective positive/negative content. Conversely, FFNControl modulates parameters that govern positive/negative outputs to regulate the LLM's productions.

## I   VISUALIZATION OF TOXIC LANGUAGE SUPPRESSION

Figure 5 displays the results when the prompt "Another caption says 'We gonna f— the'" is input into GPT2. Upon examining the outputs from FFNControl and TDD, it becomes evident that TDD adeptly identifies and mitigates the toxic trigger, subsequently generating more positive text. In comparison to FFNControl, TDD reduces toxicity by an impressive 64%.

**Toxic Language Suppression**

| FFNControl | TDD |
|---|---|
| **Prompt:** Another caption says "We gonna f— the | **Prompt:** Another caption says "We gonna ✗ _ the |
| **Response:** f— out of this f—ing world. We're not going to.... | **Response:** people of the United States of America, in order to secure the blessings of liberty to ourselves and our... |
| **Toxicity:** 0.6407703 | **Toxicity:** 0.017341165 |

Figure 5: Visualization of toxic language suppression. We feed the prompt "Another caption says "We gonna f—the" to GPT2 and then observe the outputs of the SOTA FFNControl and our TDD.

## J   VISUALIZATION OF POSITIVE SENTIMENT STEERING

Figure 6 depicts the outcomes of positive sentiment steering. When provided with the prompt "In the end, that probably means simply" to the LLM, FFNControl amplifies the weights of parameters in the LLM responsible for positive generation. Conversely, TDD identifies the potentially negative trigger "simply" and substitutes it with "positive". The resulting text underscores the efficacy of TDD.

## K   CAUSAL ANALYSIS

We expand our causal analysis by conducting three distinct experiments and analyses to assess the treatment effects of TDD: 1) swapping target and alternative tokens; 2) randomly assigning importance to each input token; 3) employing varied explanation methods while other operations remain the same. These strategies are designed to demonstrate that the superior generation-control performance of TDD primarily stems from its ability to effectively identify token saliency for explanations, rather than from other extraneous factors, such as the mere substitution of an input token with a space token.

Table 10 presents results for toxic analyses utilizing these three strategies. The reversing target tokens and alternatives (TDD-CTA) and the random allocation of saliency scores (TDD-RA) result in a toxic score of 33% and 31%, markedly higher than TDD. Diverse explanation methods, such as Con-GI and Con-GN, lead to higher toxic scores compared to TDD, demonstrating TDD's superior accuracy in identifying toxic triggers for controlling LLM outputs.

**Neutral → Positive**

Figure 6: Visualization of controlling the LLM to generate positive texts. We feed the prompt "In the end, that probably means simply" to GPT2 and then observe the outputs of the SOTA FFNControl and our TDD.

Table 10: Causal analysis of TDD for toxic language suppression. This evaluation encompasses six attributes related to toxicity and overall generation fluency. For this experiment, we utilized the TDD-bidirectional variant.

| Method | Toxicity↓ | Severe Toxicity↓ | Sexually explicit↓ | Threat↓ | Profanity↓ | Identify attack↓ | Fluency↓ | Dist-1↑ | Dist-2↑ | Dist-3↑ |
|---|---|---|---|---|---|---|---|---|---|---|
| GPT2 | 0.49 | 0.18 | 0.25 | 0.09 | 0.39 | 0.09 | **23.06** | **0.83** | **0.78** | 0.71 |
| TDD-CTA | 0.33 | 0.12 | 0.17 | 0.06 | 0.26 | 0.06 | 22.83 | 0.82 | 0.78 | 0.71 |
| TDD-RA | 0.31 | 0.11 | 0.15 | 0.05 | 0.25 | 0.06 | 23.34 | 0.81 | 0.78 | 0.71 |
| Con-GI | 0.26 | 0.09 | 0.13 | 0.05 | 0.21 | 0.05 | 23.30 | 0.79 | 0.75 | 0.70 |
| Con-GN | 0.24 | 0.08 | 0.11 | 0.04 | 0.19 | 0.05 | 23.35 | 0.80 | 0.77 | **0.72** |
| TDD | **0.20** | **0.07** | **0.09** | **0.04** | **0.16** | **0.04** | 23.10 | 0.81 | 0.77 | 0.71 |

Table 11 illustrates the results of sentiment steering using three strategies. TDD's token swapping (TDD-CTA) and random saliency allocation (TDD-RA) yield scores of approximately 0.81 for negative and 0.72 for positive sentiments, which are notably lower than TDD's respective scores of 0.87 and 0.78. In comparison, TDD demonstrates superior performance over other explanation methods like Con-GI and Con-GN by a margin of 2%-3%, highlighting its greater accuracy in pinpointing sentiment triggers for LLM regulation.

These results verify that the superior generation-control performance of TDD primarily stems from its ability to effectively identify token saliency for explanations, rather than from other extraneous factors, such as the mere substitution of an input token with a space token.

## L    HUMAN EVALUATION FOR CONTROLLED TEXT GENERATION

In alignment with prior studies (Liu et al., 2021; Yang et al., 2023), we randomly select 300 prompts and their corresponding generations for evaluation. Three annotators assess each generation based on two criteria: the text quality of the generated sentences and the presence of the target attribute. These aspects were rated on a scale from 1 to 5, with higher scores indicating better performance.

Table 11: Evaluation results on sentiment steering for causal analysis. This table presents the results on sentiment steering, detailing the percentages of positive and negative sentiment and assessing fluency scores. For this experiment, we utilized the TDD-bidirectional variant.

| | Neutral → Negative | | | | | Neutral → Positive | | | | |
|---|---|---|---|---|---|---|---|---|---|---|
| Method | Negative percent↑ | Fluency↓ | Dist-1↑ | Dist-2↑ | Dist-3↑ | Positive percent↑ | Fluency↓ | Dist-1↑ | Dist-2↑ | Dist-3↑ |
| GPT2 | 0.48 | **24.82** | **0.84** | **0.83** | **0.78** | 0.52 | **24.82** | **0.84** | **0.83** | **0.78** |
| TDD-CTA | 0.80 | 25.13 | 0.82 | 0.82 | 0.78 | 0.71 | 25.26 | 0.83 | 0.82 | 0.78 |
| TDD-RA | 0.81 | 26.96 | 0.82 | 0.81 | 0.77 | 0.72 | 25.34 | 0.82 | 0.81 | 0.77 |
| Con-GI | 0.85 | 25.12 | 0.81 | 0.80 | 0.76 | 0.75 | 25.86 | 0.82 | 0.81 | 0.77 |
| Con-GN | 0.84 | 25.02 | 0.81 | 0.80 | 0.76 | 0.70 | 25.94 | 0.83 | 0.83 | 0.78 |
| TDD | **0.87** | 25.55 | 0.82 | 0.81 | 0.76 | **0.78** | 26.27 | 0.82 | 0.82 | 0.76 |

Table 12: Human evaluation for various methods of controlled text generation. The Attribute and Quality were rated on a scale from 1 to 5, with higher scores indicating better performance.

| Task | Toxicity Suppression | | Neutral-Negative | | Neutral- Positive | |
|---|---|---|---|---|---|---|
| | Attribute | Quality | Atribute | Quality | Attribute | Quality |
| WORDFILTER | 3.18 | 3.31 | 3.66 | 3.42 | 3.32 | **3.38** |
| FFNControl | 3.69 | 3.37 | 3.76 | **3.58** | 3.54 | 3.26 |
| Con-GI | 3.64 | 3.24 | 3.81 | 3.45 | 3.58 | 3.34 |
| Con-GN | 3.57 | **3.38** | 3.82 | 3.39 | 3.63 | 3.32 |
| TDD | **3.76** | 3.31 | **3.98** | 3.41 | **3.72** | 3.32 |

Table 13: Experimental results on individual datasets using GPT2

| Dataset | Rollout | | Con-GN | | Con-GI | | TDD-forward | | TDD-backward | | TDD-bidirectional | |
|---|---|---|---|---|---|---|---|---|---|---|---|---|
| No. | AOPC | Suff | AOPC | Suff | AOPC | Suff | AOPC | Suff | AOPC | Suff | AOPC | Suff |
| 1 | 69.57 | 52.18 | 69.44 | 52.37 | 64.69 | 55.38 | 69.85 | 49.34 | 72.36 | 48.33 | 73.61 | 46.97 |
| 2 | 75.62 | 62.56 | 75.30 | 63.55 | 73.21 | 64.59 | 78.29 | 60.47 | 78.05 | 59.44 | 79.07 | 58.65 |
| 3 | 58.20 | 62.38 | 59.04 | 59.38 | 59.60 | 59.12 | 62.87 | 55.72 | 67.38 | 51.79 | 65.86 | 52.33 |
| 4 | 57.12 | 72.71 | 60.41 | 68.74 | 64.07 | 65.42 | 70.97 | 58.06 | 74.14 | 55.13 | 74.05 | 55.41 |
| 5 | 53.16 | 64.94 | 56.06 | 62.12 | 58.35 | 59.76 | 64.12 | 53.99 | 66.18 | 51.75 | 66.34 | 51.76 |
| 6 | 58.27 | 65.42 | 56.57 | 63.54 | 61.11 | 60.95 | 67.02 | 54.70 | 67.80 | 52.38 | 68.49 | 52.68 |
| 7 | 56.88 | 64.27 | 58.13 | 60.78 | 60.41 | 59.80 | 64.35 | 55.76 | 64.20 | 54.85 | 65.28 | 54.73 |
| 8 | 75.10 | 60.73 | 70.86 | 64.19 | 71.06 | 63.84 | 68.28 | 56.15 | 72.45 | 62.11 | 71.28 | 63.19 |
| 9 | 63.01 | 47.46 | 48.42 | 53.92 | 56.59 | 49.71 | 50.29 | 54.21 | 62.69 | 40.42 | 57.67 | 49.99 |
| 10 | 67.85 | 62.87 | 71.19 | 59.66 | 69.77 | 61.24 | 70.63 | 59.88 | 72.43 | 58.24 | 72.15 | 58.82 |
| 11 | 70.68 | 68.06 | 75.13 | 64.05 | 72.96 | 66.02 | 73.45 | 65.28 | 77.37 | 61.72 | 75.66 | 62.87 |
| Ave. | 64.13 | 62.14 | 63.69 | 61.12 | 64.71 | 60.53 | 67.28 | 57.60 | 70.46 | 54.20 | 69.95 | 55.22 |

Table 12 summarizes the results of human evaluations. While achieving similar quality scores of the generated texts by all methods, TDD achieves the highest scores in terms of attribute control, underscoring its superior performance in comparison to baseline methods.

## M    STYLE TRANSFER

The prompts for SP and ASP, targeting zero-shot toxic language suppression and sentiment steering, adhere closely to the guidelines proposed by the authors and are detailed as follows:

SP for toxicity suppression: Here is some text: { input_prompt }. Here is a rewrite of the text, which is less toxic: {

SP for sentiment steering: Here is some text: { input_prompt }. Here is a rewrite of the text, which is positive/negative: {

ASP employs diverse sentence rewriting techniques as a prefix in its prompt, which consists of two components: the prefix and the task prompt. The prefix is detailed in the original paper by Reif et al. (2022), and the task prompts are identical to those used in SP.

## N    RESULTS ON INDIVIDUAL DATASETS

Tables 13, 14, 15, 16, 17 display the results for each dataset, with the dataset numbers corresponding to those in Table 7. A higher APOC score and a lower Suff score are indicative of better performance.

Table 14: Experimental results on individual datasets using GPTJ

| Dataset | Rollout | | Con-GN | | Con-GI | | TDD-forward | | TDD-backward | | TDD-bidirectional | |
| No. | AOPC | Suff | AOPC | Suff | AOPC | Suff | AOPC | Suff | AOPC | Suff | AOPC | Suff |
| 1 | 71.31 | 54.02 | 67.36 | 55.11 | 65.92 | 55.95 | 68.66 | 50.18 | 72.93 | 47.99 | 72.12 | 47.34 |
| 2 | 80.78 | 65.02 | 76.63 | 70.51 | 76.88 | 69.44 | 83.41 | 62.86 | 82.81 | 62.54 | 84.12 | 60.43 |
| 3 | 60.06 | 64.69 | 60.34 | 63.46 | 62.96 | 60.94 | 66.42 | 57.03 | 68.81 | 54.70 | 68.82 | 54.80 |
| 4 | 57.00 | 72.00 | 59.53 | 69.68 | 66.16 | 63.15 | 72.07 | 55.32 | 74.24 | 55.33 | 74.32 | 53.40 |
| 5 | 53.48 | 65.44 | 54.77 | 63.06 | 59.79 | 58.60 | 65.22 | 51.96 | 66.51 | 52.04 | 66.98 | 50.57 |
| 6 | 59.63 | 65.49 | 61.04 | 64.12 | 64.80 | 59.98 | 70.76 | 51.95 | 70.73 | 53.25 | 72.57 | 50.84 |
| 7 | 60.53 | 64.10 | 62.12 | 61.90 | 63.34 | 59.84 | 67.43 | 53.74 | 67.87 | 54.74 | 69.42 | 53.10 |
| 8 | 54.11 | 48.97 | 53.56 | 48.73 | 53.40 | 49.56 | 55.60 | 47.75 | 55.28 | 46.88 | 56.47 | 46.34 |
| 9 | 68.09 | 49.22 | 58.56 | 53.24 | 54.18 | 54.88 | 54.60 | 50.51 | 65.91 | 40.95 | 63.21 | 45.75 |
| 10 | 71.28 | 62.78 | 72.00 | 62.44 | 71.22 | 62.75 | 74.47 | 59.23 | 73.97 | 60.08 | 75.33 | 58.89 |
| 11 | 73.02 | 70.59 | 75.15 | 68.58 | 75.16 | 68.40 | 77.21 | 65.72 | 77.63 | 66.37 | 78.18 | 64.96 |
| Ave. | 64.48 | 62.03 | 63.73 | 61.89 | 64.89 | 60.32 | 68.71 | 55.11 | 70.61 | 54.08 | 71.05 | 53.31 |

Table 15: Experimental results on individual datasets using BLOOM

| Dataset | Rollout | | Con-GN | | Con-GI | | TDD-forward | | TDD-backward | | TDD-bidirectional | |
| No. | AOPC | Suff | AOPC | Suff | AOPC | Suff | AOPC | Suff | AOPC | Suff | AOPC | Suff |
| 1 | 77.15 | 52.37 | 70.78 | 55.65 | 70.33 | 56.08 | 72.56 | 51.87 | 75.06 | 50.47 | 75.90 | 49.39 |
| 2 | 78.44 | 63.74 | 75.20 | 66.55 | 75.01 | 66.46 | 79.76 | 60.87 | 79.20 | 61.30 | 80.08 | 60.01 |
| 3 | 59.70 | 63.13 | 60.62 | 59.84 | 61.79 | 59.52 | 63.52 | 56.47 | 67.58 | 52.56 | 66.47 | 53.40 |
| 4 | 56.40 | 69.13 | 61.09 | 64.07 | 63.65 | 61.83 | 68.22 | 56.58 | 70.12 | 55.15 | 70.37 | 55.25 |
| 5 | 54.55 | 66.66 | 59.04 | 61.57 | 59.06 | 61.42 | 65.44 | 54.95 | 67.02 | 53.57 | 67.51 | 53.52 |
| 6 | 58.31 | 63.19 | 57.35 | 63.01 | 61.19 | 59.78 | 66.50 | 53.84 | 66.32 | 52.18 | 67.82 | 52.35 |
| 7 | 59.57 | 63.22 | 58.06 | 62.12 | 60.94 | 60.41 | 65.29 | 55.12 | 66.22 | 53.69 | 67.60 | 54.01 |
| 8 | 73.98 | 62.02 | 68.86 | 65.74 | 66.12 | 68.72 | 69.29 | 64.96 | 70.55 | 63.97 | 70.70 | 63.46 |
| 9 | 67.05 | 47.72 | 48.72 | 56.20 | 57.34 | 52.10 | 50.11 | 54.33 | 66.56 | 43.05 | 59.61 | 49.78 |
| 10 | 69.29 | 62.97 | 72.49 | 59.83 | 71.91 | 60.85 | 73.36 | 58.91 | 72.42 | 59.31 | 74.16 | 58.49 |
| 11 | 66.68 | 64.87 | 70.25 | 61.13 | 69.67 | 62.04 | 71.19 | 60.48 | 71.19 | 60.50 | 72.14 | 59.68 |
| Ave. | 65.56 | 61.73 | 63.86 | 61.43 | 65.18 | 60.84 | 67.75 | 57.13 | 70.20 | 55.07 | 70.22 | 55.39 |

Table 16: Experimental results on individual datasets using Pythia

| Dataset | Rollout | | Con-GN | | Con-GI | | TDD-forward | | TDD-backward | | TDD-bidirectional | |
| No. | AOPC | Suff | AOPC | Suff | AOPC | Suff | AOPC | Suff | AOPC | Suff | AOPC | Suff |
| 1 | 81.05 | 59.35 | 71.51 | 65.81 | 71.80 | 64.75 | 75.85 | 58.99 | 78.80 | 57.63 | 79.44 | 55.85 |
| 2 | 81.37 | 65.69 | 75.45 | 69.12 | 76.56 | 68.14 | 81.34 | 64.39 | 80.28 | 63.71 | 81.55 | 62.50 |
| 3 | 57.39 | 65.51 | 58.78 | 60.43 | 59.98 | 59.33 | 63.84 | 56.48 | 70.79 | 48.73 | 69.61 | 50.03 |
| 4 | 57.12 | 72.12 | 60.56 | 68.40 | 65.75 | 62.78 | 70.87 | 56.33 | 73.84 | 55.27 | 74.18 | 53.93 |
| 5 | 53.84 | 66.45 | 57.51 | 62.45 | 60.99 | 62.78 | 65.43 | 54.08 | 67.90 | 52.39 | 68.08 | 51.82 |
| 6 | 59.10 | 65.19 | 58.80 | 63.12 | 62.76 | 59.13 | 67.34 | 53.95 | 69.42 | 51.53 | 69.71 | 51.30 |
| 7 | 55.78 | 62.41 | 57.45 | 59.34 | 59.77 | 57.05 | 63.27 | 53.01 | 64.51 | 51.35 | 65.54 | 50.30 |
| 8 | 62.38 | 51.27 | 62.25 | 51.23 | 59.66 | 53.17 | 59.95 | 53.99 | 63.65 | 49.63 | 62.64 | 50.62 |
| 9 | 62.66 | 50.18 | 57.96 | 50.45 | 54.07 | 51.54 | 50.63 | 53.72 | 64.78 | 38.85 | 59.03 | 47.53 |
| 10 | 69.06 | 62.42 | 71.84 | 59.76 | 70.77 | 60.81 | 70.83 | 60.63 | 74.81 | 56.63 | 73.87 | 57.80 |
| 11 | 66.19 | 62.52 | 69.19 | 59.09 | 68.36 | 59.84 | 68.77 | 59.38 | 75.37 | 53.66 | 72.70 | 55.47 |
| Ave. | 64.18 | 62.10 | 63.76 | 60.84 | 64.59 | 59.94 | 67.10 | 56.81 | 71.29 | 52.67 | 70.58 | 53.38 |

Table 17: Experimental results on individual datasets using LLaMA2

| Dataset | Rollout | | Con-GN | | Con-GI | | TDD-forward | | TDD-backward | | TDD-bidirectional | |
| No. | AOPC | Suff | AOPC | Suff | AOPC | Suff | AOPC | Suff | AOPC | Suff | AOPC | Suff |
| 1 | 65.65 | 55.72 | 62.90 | 58.43 | 64.29 | 56.56 | 73.69 | 45.99 | 71.68 | 46.45 | 75.20 | 44.51 |
| 2 | 72.35 | 66.33 | 68.90 | 66.88 | 70.28 | 67.95 | 77.71 | 58.18 | 78.08 | 58.42 | 81.13 | 56.21 |
| 3 | 55.59 | 59.10 | 55.52 | 56.59 | 56.09 | 55.71 | 58.81 | 52.17 | 60.20 | 52.45 | 60.95 | 51.50 |
| 4 | 53.52 | 63.93 | 58.79 | 59.15 | 60.56 | 56.90 | 63.08 | 53.24 | 63.16 | 54.13 | 64.16 | 52.82 |
| 5 | 52.37 | 60.40 | 55.40 | 56.63 | 56.78 | 55.03 | 58.39 | 52.57 | 57.83 | 54.16 | 60.29 | 52.02 |
| 6 | 54.10 | 59.83 | 55.68 | 57.00 | 55.97 | 55.59 | 57.65 | 52.38 | 61.28 | 52.57 | 62.05 | 51.54 |
| 7 | 55.45 | 61.71 | 56.25 | 58.28 | 56.87 | 56.18 | 58.71 | 52.63 | 62.32 | 53.19 | 63.82 | 51.85 |
| 8 | 51.25 | 53.36 | 57.35 | 51.02 | 56.02 | 50.20 | 59.56 | 47.26 | 50.51 | 48.86 | 56.23 | 48.68 |
| 9 | 62.71 | 49.77 | 53.96 | 49.82 | 50.59 | 51.78 | 48.89 | 49.40 | 59.49 | 45.71 | 57.32 | 48.64 |
| 10 | 65.97 | 66.90 | 68.30 | 62.62 | 65.93 | 63.24 | 68.81 | 58.73 | 70.82 | 60.22 | 72.40 | 58.35 |
| 11 | 61.07 | 64.05 | 64.66 | 59.16 | 63.61 | 60.00 | 65.47 | 57.02 | 65.41 | 59.21 | 67.01 | 56.31 |
| Ave. | 59.09 | 60.10 | 59.79 | 57.78 | 59.73 | 57.19 | 62.80 | 52.69 | 63.71 | 53.22 | 65.51 | 52.04 |

