# OpenReview forum: "Unveiling and Manipulating Prompt Influence in Large Language Models"
_ICLR.cc/2024/Conference — ICLR 2024 poster_

### Official Review · Reviewer_rNoS · 2023-10-28

**Soundness:** 3 good
**Presentation:** 3 good
**Contribution:** 2 fair
**Rating:** 6
**Confidence:** 4

**Summary:**

Existing saliency methods have problems that are inconsistent with LLM generation goals or rely heavily on linear assumptions. To address this, this paper proposes token distribution dynamics (TDD) to unveil and manipulate the role of prompts in generating LLM outputs. TDD leverages the interpreting capabilities of the language model head to assess input saliency. It projects input tokens into the embedding space and then estimates their significance based on distribution dynamics over the vocabulary. Experiments reveal that the TDD surpasses baselines in elucidating the causal relationships between prompts and LLM outputs and achieves good results on two prompt manipulation tasks for control text generation tasks.

**Strengths:**

1. The motivation of this paper is clear, and the method is simple and effective. The proposed token distribution dynamics (TDD) can unveil and manipulate the role of prompts in generating LLM outputs and it simply depends on the dynamic of token distribution.
2. The method proposed in this paper works well and has achieved improved results on 11 datasets.
3. In addition to achieving great improvement in elucidating the causal relationship between prompts and LLM output, the method in this paper also demonstrates its potential application value in controllable text generation tasks.

**Weaknesses:**

1. The TDD method is based on multiple assumptions listed at the bottom of page 4. Some assumptions are strong without any theoretical and experimental proof, especially the first and second assumptions.
2. For controllable text generation, this paper uses simple meaningless space tokens as alternative tokens in toxic language mitigation. In sentiment direction, this paper uses simple key tokens “positive” or “negative” as alternative tokens. Although this can prove that the method in this paper has potential application value in controllable text generation tasks, this method is too simple for practical applications. And whether such a way will impact the reasonableness of the evaluation of the results.
3. In controllable text generation tasks, only automatic evaluation is used, manual evaluation is lacking, and the indicators used in automatic evaluation are relatively simple. It is difficult to fully reflect the quality of the generated text, especially when there is no ground truth for the task of this paper.
4. LLMs can already complete controllable text generation tasks well, such as by giving LLMs toxic prefixes and designed prompts, making LLMs generate non-toxic text. This paper lacks a comparison with them, which may reduce the application value of this work.
5. The attribution of the prompt for the next token prediction is a simple problem that has been explored in many text classification tasks. The SOTA method such as integrated gradients (IG) should be also considered as a baseline. Furthermore, I think the attribution of the prompt for the sequence of generated text is a more practical task than the current one.

**Questions:**

The major concerns are listed in the Weaknesses.

Some other questions are listed below.

1. If it is in controllable text generation under multi-attribute control, what is the scalability and performance of this method?
2. In the task of controllable text generation, the data used in this paper does not have ground truth, and the existing evaluation results cannot fully reflect the quality of the generated text. How does the method in this paper perform under manual evaluation?
3. The highlighted number in Table 3 column 5 is wrong.

---

> ### Author Response · Authors · 2023-11-16
> **Response to Reviewer rNoS (1/4)**
>
> Dear Reviewer rNoS,
>
> Thank you for your detailed and constructive feedback on our paper. We have carefully considered your insights and have addressed the identified weaknesses (**W**) as well as responded to the questions (**Q**) you raised. Our comprehensive responses (**A**) are provided below for your review.
>
> [**W1**] The TDD method is based on multiple assumptions listed at the bottom of page 4. Some assumptions are strong without any theoretical and experimental proof, especially the first and second assumptions.
>
> [**Q1**] We are sorry for the insufficient clarity in articulating the theoretical and empirical foundations for the assumptions presented on page 4. In response, we reassess the assumptions underlying TDD from both theoretical and empirical perspectives.
>
> **Theoretical Justification**
>
> Geva et al. (2021; 2022) have theoretically demonstrated that in GPT2, token representations at each layer can be projected as evolving distributions over the vocabulary. Considering the consistent decoder-only transformer architecture in contemporary autoregressive language models (LLMs) such as GPTJ and LLaMA2, this theoretical analysis is extendable to these models as well since their mathematical derivations are exactly the same. This supports our first assumption/ finding: the concept of “token representations at each layer as evolving distributions” is applicable across various LLMs.
> However, their empirical study only focuses on GPT2. To validate the generalization empirically, we conducted experiments across various LLMs and datasets.
>
> **Empirical Justification**
>
> Detailed descriptions of the experiments and their settings are provided in Appendix A.
>
> 1) Finding 1: The theory of “token representations at each layer as evolving distributions over the vocabulary” can be generalized across various LLMs.
> Given an input text, we first visualize the token distributions at each layer projected by the LM head for various LLMs including GPT2, GPTJ, BLOOM, Pythia and LLaMA2 in Figure 2 in Appendix A. Using LLaMA2 in Figure 2e as an example, for the initial input “ Some”, intermediate layers produce outputs such as “Someone” and “Somewhere”. With the input “Some boys discover”, outputs like “Some boys discover themselves” emerge. In the upper layers, the prediction logit incrementally rises, converging to the final prediction. These visualizations underscore that token representations from any layer can be depicted as meaningful token distributions over the vocabulary. Furthermore, this notation can be applied to all input tokens, instead of the only last token for prediction.
>
> To quantitatively assess the dynamics of token distributions across layers, we treat the token distribution at the final layer as the “ground-truth”. We then compute the KL-divergence between intermediate layer token distributions and this ground-truth to study their convergence patterns. We employ datasets from BLiMP to estimate the KL-divergence for each token. Figure 2f depicts the mean divergence across layers for all input tokens in all datasets. It can be observed that the learning process reflects the evolving token distributions over the vocabulary, which converge in a roughly monotonic manner to the distribution of the final layer.
>
> Both qualitative and quantitative analyses corroborate our first finding regarding the generalizability of token representation theory across various LLMs.
>
> 2) Finding 2: Projected token distributions from the LM head hold the potential to elucidate causal relationships between prompts and LLM outputs.
> Our first finding confirms that the LM head projects complex token representations at each layer as distributions over the vocabulary. Advancing this concept, we observe that these distributions are both interpretable and integral to the model's generative process. Each dimension of the token distribution represents a specific token in the vocabulary, with the value of each logit indicating the likelihood of the subsequent token based on current and preceding tokens. This capability of the LLM to track prediction or distribution changes with the progressive introduction of tokens enables us to infer which tokens influence the prediction of a target token, offering explanatory insights. Thus, our second finding is that projected token distributions from the LM head hold the potential to elucidate causal relationships between prompts and LLM outputs.
>
> The experiments detailed in Section 4 further validate our assumptions and findings. These experimental results show that utilizing the token distributions projected by the LM head yields the most faithful explanations. This substantiates our assertion that the LM head possesses the potential to clarify causal relationships between prompts and LLM outputs.
>
> Detailed analysis and experiments are provided in Section 3.2 and Appendix A in our revised paper.

---

> > ### Author Response · Authors · 2023-11-16
> > **Response to Reviewer rNoS (2/4)**
> >
> > [**W2**] For controllable text generation, this paper uses simple meaningless space tokens as alternative tokens in toxic language mitigation. In sentiment direction, this paper uses simple key tokens “positive” or “negative” as alternative tokens. Although this can prove that the method in this paper has potential application value in controllable text generation tasks, this method is too simple for practical applications. And whether such a way will impact the reasonableness of the evaluation of the results.
> >
> > [**A2**] Thanks for highlighting your concerns. We would like to address your concerns by clarifying three key points:
> >
> > 1) Our scenario is **zero-shot** controlled text generation, distinct from general controlled text generation. The zero-shot setting is complex yet common in real-life applications. In scenarios with limited data and GPU availability, our method offers a practical solution by treating LLMs as black-box models without requiring training samples or extensive computational resources. TDD is particularly effective in zero-shot contexts, and it significantly outperforms other baselines, demonstrating its practicality and effectiveness.
> >
> > 2) To ensure reasonableness, we have added comprehensive causal analyses. These include experiments like swapping target and alternative tokens, randomly assigning importance to tokens, and applying different explanation methods while maintaining consistency in other operations. These experimental results, detailed in Appendix K, confirm that TDD's enhanced control over text generation primarily stems from its ability to accurately identify token saliency, which is in line with our primary contribution of this paper: taking the token distribution as a new medium to estimate token saliency.
> >
> > 3) The controlled text generation experiments are designed to evaluate TDD's ability to identify key words in input prompts that influence LLM outputs. The main contributions of this paper are threefold: firstly, introducing token distributions as a novel medium for assessing token saliency; secondly, proposing TDD as a technique to leverage token distribution dynamics for understanding prompt influence on LLM generations; thirdly, demonstrating two practical applications of TDD - zero-shot toxicity suppression and sentiment steering. Our findings in Section 5 confirm TDD's efficacy in pinpointing pivotal tokens in prompts for manipulating LLM outputs in zero-shot scenarios.
> >
> > Consequently, we believe that the contribution of TDD is significant, addressing a challenging yet common application scenario. We deeply value your feedback and, in response, are committed to further refining TDD for a wider range of applications and more complex scenarios in our future work.
> >
> > [**W3**] In controllable text generation tasks, only automatic evaluation is used, manual evaluation is lacking, and the indicators used in automatic evaluation are relatively simple. It is difficult to fully reflect the quality of the generated text, especially when there is no ground truth for the task of this paper.
> >
> > [**A3**] We apologize for excluding manual evaluations. In the revised version of our paper, we have incorporated manual evaluation metrics and results, as detailed below.
> >
> > In alignment with prior studies (Liu et al., 2021; Yang et al., 2023), we randomly selected 300 prompts and their corresponding generations for evaluation. Three annotators assessed each generation based on two criteria: the text quality of the generated sentences and the presence of the target attribute. These aspects were rated on a scale from 1 to 5, with higher scores indicating better performance.
> >
> > Table 12 in Appendix L summarizes the results of human evaluations. While achieving similar quality scores of the generated texts by all methods, TDD achieves the highest scores in terms of attribute control, underscoring its superior performance in comparison to baseline methods.
> >
> > This revision is provided in Section 5.3 and Appendix L.

---

> > > ### Author Response · Authors · 2023-11-16
> > > **Response to Reviewer rNoS (3/4)**
> > >
> > > [**W4**] LLMs can already complete controllable text generation tasks well, such as by giving LLMs toxic prefixes and designed prompts, making LLMs generate non-toxic text. This paper lacks a comparison with them, which may reduce the application value of this work.
> > >
> > > [**A4**] Thank you for highlighting this oversight. We apologize for not previously providing detailed justifications for our baseline selections. In response, we have expanded our comparison to include a broader range of baselines.
> > >
> > > 1) While prefix-tuning is a common approach for controlled text generation, it necessitates training samples to train the prefix tokens. This requirement does not align with our focus on zero-shot controlled text generation. Hence, we opted not to include baselines derived from prefix-tuning methods in our analysis.
> > >
> > > 2) Designed prompts for controlled text generation such as prompting for style transfer can be utilized in the zero-shot scenario. We employ two methods: 1) Style Prompting (SP), which integrates specific style requirements, such as toxicity and sentiment, directly into the input prompts; 2) augmented Style Prompting (ASP), which serves as a formidable baseline by prompting the model with various examples of sentence rewriting techniques, thereby facilitating a range of sentence rewriting tasks. Detailed information regarding these two prompting strategies can be found in Section 5.1 and Appendix M.
> > >
> > >
> > > The efficacy of these methods in zero-shot toxic language suppression and sentiment manipulation is presented in Tables 3 and 4 in our revised paper. These results reveal that SP and ASP are less effective compared to other state-of-the-art baselines like WORDFILTER and FFNControl. This indicates models like GPT-2 struggle with comprehending these prompts, although they may be more effective for larger LLMs such as GPT-4.
> > >
> > > This revision is provided in Sections 5.1 and 5.2.
> > >
> > > [**W5**] The attribution of the prompt for the next token prediction is a simple problem that has been explored in many text classification tasks. The SOTA method such as integrated gradients (IG) should be also considered as a baseline. Furthermore, I think the attribution of the prompt for the sequence of generated text is a more practical task than the current one.
> > >
> > > [**A5**] Thanks for your valuable feedback. In response, we have incorporated additional SOTA baselines, including Gradient * Input (GI) and Gradient Norms (GN), into our analysis. The experimental results, summarized in Table 2, reveal that both GI and GN are outperformed by their contrastive versions Con-GI and Con-GN. Importantly, our proposed TDD method continues to significantly surpass all these baselines in performance.
> > >
> > > To the best of our knowledge, we are the first to explore token distribution as a medium to estimate token saliency. TDD can be applied to various scenarios including 1) a single target token and a single alternative token, 2) only target tokens, 3) multiple target tokens and multiple alternative tokens, and 4) controlled attributes in generated sequences. More details can be found in Appendix B.2 in our revised paper.
> > >
> > > Therefore, while the application of TDD to entire sequences has not been extensively discussed or explored in this paper, we have provided a thorough analysis and conducted detailed experiments at the token level across various scenarios. We are grateful for your insightful feedback and plan to investigate the application of TDD to sequence-level explanations in our future research endeavors.

---

> > > > ### Author Response · Authors · 2023-11-16
> > > > **Response to Reviewer rNoS (4/4)**
> > > >
> > > > [**Q1**] If it is in controllable text generation under multi-attribute control, what is the scalability and performance of this method?
> > > >
> > > > [**A6**] Thank you for raising this important question. Our current research is focused on zero-shot controlled text generation, specifically targeting single-attribute control. Multi-attribute control, indeed, presents a more complex challenge, primarily due to the increased dimensionality and potential conflicts between attributes. While our current study does not extend to this area, we recognize its significance in the broader context of text generation. In future work, we aim to explore strategies such as hierarchical attribute control or advanced learning algorithms to manage the intricacies of multi-attribute control. These approaches could potentially enhance scalability and improve performance in handling multiple attributes simultaneously, which we believe is a valuable direction for the evolution of text generation technologies.
> > > >
> > > > [**Q2**] In the task of controllable text generation, the data used in this paper does not have ground truth, and the existing evaluation results cannot fully reflect the quality of the generated text. How does the method in this paper perform under manual evaluation?
> > > >
> > > > [**A7**] Please refer to our response [A3] to weakness 3 [W3].
> > > >
> > > >
> > > > [**Q3**] The highlighted number in Table 3 column 5 is wrong.
> > > >
> > > > [**A8**] We appreciate your attention to detail in identifying this error. We have rectified the incorrect highlight in Table 3, column 5, and have conducted a comprehensive review of all other tables in our manuscript to verify their accuracy. This additional scrutiny ensures that all data is presented correctly and reliably. We apologize for any inconvenience this mistake may have caused and thank you for your contribution to improving the quality of our work.
> > > >
> > > > **References**
> > > >
> > > > Mor Geva, Roei Schuster, Jonathan Berant, and Omer Levy. Transformer feed-forward layers are key-value memories.
> > > >
> > > > Mor Geva, Avi Caciularu, Kevin Wang, and Yoav Goldberg. Transformer feed-forward layers build predictions by promoting concepts in the vocabulary space.
> > > >
> > > > Alisa Liu, Maarten Sap, Ximing Lu, Swabha Swayamdipta, Chandra Bhagavatula, Noah A. Smith, and Yejin Choi. DExperts: Decoding-time controlled text generation with experts and anti- experts.
> > > >
> > > > Kexin Yang, Dayiheng Liu, Wenqiang Lei, Baosong Yang, Mingfeng Xue, Boxing Chen, and Jun Xie. Tailor: A soft-prompt-based approach to attribute-based controlled text generation.

---

> > > > > ### Comment · Reviewer_rNoS · 2023-11-21
> > > > >
> > > > > Thanks for the detailed response, while some concerns are still there, I will not modify my score.
> > > > >
> > > > > - A5 as I have mentioned the SOTA is integrated gradients (IG) but the author does not consider this baseline.
> > > > > - there is no definition of 'zero-shot controlled text generation' in the paper and response, what is the meaning of zero-shot? Does that mean other baselines are not zero-shot?

---

> ### Author Response · Authors · 2023-11-22
> **Response to Reviewer rNoS**
>
> Dear Reviewer rNoS,
>
> Thank you for your follow-up. We have further addressed your concerns as outlined below:
>
> **1**. A5 as I have mentioned the SOTA is integrated gradients (IG) but the author does not consider this baseline.
>
> We apologize for this omission in our first revision. We have recently completed the experiments involving IG and have now included these results in our revised paper. Here are the details:
>
> In addition to our proposed method, we have compared it with the advanced explanation method Integrated Gradients (IG) (Sundararajan et al., 2017). The experimental results, presented in Table 2 of our revised paper, show that IG slightly underperforms compared to the SOTA contrastive method Con-GI. However, our TDD significantly outperforms all baselines by a considerable margin of 5%-7%. We argue that the suboptimal performance of IG is primarily due to its focus solely on target tokens while disregarding alternative tokens.
>
> These revisions and detailed results can be found in Sections 4.4-4.5 and Table 2 of our revised paper.
>
> **2** there is no definition of 'zero-shot controlled text generation' in the paper and response, what is the meaning of zero-shot? Does that mean other baselines are not zero-shot?
>
> We apologize for not previously defining “zero-shot controlled text generation.” This term refers to a scenario where no training samples are available or provided, and all methods must be executed without such samples.
>
> Our TDD approach is designed to function without any training samples. To ensure a fair comparison, we selected six baselines that are also applicable in a zero-shot setting. This setting, common yet challenging in real-life applications, tests the ability to control attributes like sentiment and toxicity in large language models (LLMs) without prior training. Our experiments demonstrate that TDD effectively manages these attributes under zero-shot conditions.
>
> We hope our response resolves your concerns. Please do not hesitate to contact us if you have further questions.

---

> > ### Comment · Reviewer_rNoS · 2023-11-22
> >
> > It sounds reasonable and addresses most of my concerns. I plan to raise my score.
> >
> > Further small suggestions, 1) the colors and alignment of Figure 1 should be improved; 2) there are many text style transfer works that should also be mentioned in this paper.
> >
> > For example, papers saying "unsupervised text style transfer" are somehow similar to the application,
> > [1] Unsupervised text style transfer using language models as discriminators
> > [2] Transductive learning for unsupervised text style transfer
> > [3] Reformulating unsupervised style transfer as paraphrase generation

---

> > > ### Author Response · Authors · 2023-11-22
> > > **Response to Reviewer rNoS**
> > >
> > > Dear Reviewer rNoS,
> > >
> > > Thank you very much for your thorough review. We are pleased to learn that most of your concerns have been addressed.
> > >
> > > We also appreciate your kind suggestions regarding Figure 1 and the inclusion of additional related works on text style transfer. We will make these modifications promptly and incorporate them into our revised paper

---

> > > > ### Author Response · Authors · 2023-11-22
> > > > **Response to Reviewer rNoS - updated manuscript**
> > > >
> > > > Dear Reviewer rNoS,
> > > >
> > > > Thank you once more for your valuable suggestions. We have updated the manuscript accordingly, revising Figure 1 and including additional references on style transfer. The revised version has been uploaded.
> > > >
> > > > Should you have any further questions or require additional information, please feel free to contact us.

---

### Official Review · Reviewer_jvM3 · 2023-10-29

**Soundness:** 3 good
**Presentation:** 2 fair
**Contribution:** 2 fair
**Rating:** 6
**Confidence:** 4

**Summary:**

This paper considers the roles of individual tokens in prompts in guiding the responses of LLMs. To analyze the dynamics of token distributions, this paper proposes three methods (TDD-forward, TDD-backward, TDD-bidirectional), each offering insights into token relevance.

TDD-forward computes $r_i - r_{i-1}$, where each $r_i$ is the difference between the probability of token i (computed from a contextualized LM) and the probability of the alternative token. TDD-backward computes the probabilities at the last token, and uses $r_{i-1} - r_i$. TDD-bidirectional is the sum of the TDD-forward and TDD-backward.

On 11 out of the 67 datasets on BLiMP, this paper shows that TDD-based methods in general have better AOPC and Sufficiency results than multiple baseline methods. Additionally, in some case studies (toxic language suppression and controllable text generation), the TDD method works better than some baseline methods.

**Strengths:**

- The method is simple and efficient.
- The analysis towards understanding causal mechanisms of token probability distributions is a timely and important research topic.

**Weaknesses:**

- The TDD methods heavily depend on the alternative word $w_a$. The datasets chosen in this paper (those BLiMP datasets) provide sentence pairs with exactly one-word differences. Other datasets may not have such well-defined alternative words. This greatly limits the potential applicability of the proposed TDD methods.
    - A related point: it is unclear to me how the w_a in the Section 5 experiments are identified.
- The analyses presented in this paper are not really causal analyses, despite claims that the LM head “elucidate causal relationships between prompts and LLM outputs” (page 4). No controlled variable is specified, and no treatment effect is measured. This paper could be much stronger if some sort of causal scores are computed, for example, how large is the treatment effect of changing w to w_a causes the toxicity of the generated texts. Currently, the numbers shown in Tables 3 and 4 look similar to the treatment effects, but without more descriptions about how the variables are controlled, we can’t really say the numbers are treatment effects.
- A minor point about writing styles: Excessive adverbs (e.g., those in the sentence “elegantly simple yet remarkably effective”) can make the paper read less rigorous, rendering the text resemble more of a social media post than a scientific paper.

**Questions:**

In section 5, which variant of TDD is used? Are TDD-forward used there?

---

> ### Author Response · Authors · 2023-11-16
> **Response to Reviewer jvM3 (1/2)**
>
> Dear Reviewer jvM3,
>
> We appreciate your comprehensive review and valuable feedback on our manuscript. In response, we have systematically addressed the identified weaknesses (**W**) and answered your queries (**Q**). Our detailed responses (**A**) are outlined below.
>
> [**Q1**] In section 5, which variant of TDD is used? Are TDD-forward used there?
>
> [**A1**] We apologize for the lack of clarity regarding our choice of TDD variants for the controlled text generation. In our application experiments in section 5, we specifically chose the TDD-bidirectional variant for both toxicity suppression and sentiment steering. This decision was based on its superior performance observed in our main experiments, which are thoroughly discussed in Section 4. We have added such clarification in our revised manuscript.
>
> [**W1**] The TDD methods heavily depend on the alternative word w_a. The datasets chosen in this paper (those BLiMP datasets) provide sentence pairs with exactly one-word differences. Other datasets may not have such well-defined alternative words. This greatly limits the potential applicability of the proposed TDD methods.  A related point: it is unclear to me how the w_a in the Section 5 experiments are identified.
>
> [**A2**] Thank you for highlighting your concerns. We apologize for any lack of clarity regarding the scope of TDD in our paper. Actually, **TDD's effectiveness is not strictly dependent on alternative tokens. TDD is versatile and can be applied in various scenarios, including those with 1) a single target token and a single alternative token, 2) only target tokens, 3) multiple target tokens and multiple alternative tokens, and 4) controlled attributes in generated sequences.** We substantiate each of these applications with experimental evidence as follows:
>
> **Scope of TDD**
>
> 1) Single Target and Single Alternative Token: Section 4 details our main experiments, which validate TDD's effectiveness in scenarios involving a single target token and a single alternative token.
>
> 2) Target Token Only: TDD is applicable even in the absence of alternative tokens. By assigning a zero probability to alternative tokens in Equations 2 and 4 in the paper, we can negate the necessity of alternative tokens. In the added experiments, we estimate each token’s importance by using only target tokens, and the alternative tokens are not provided. The results, summarized in Table 5, show TDD's significant outperformance of strong baselines—by about 3%—in scenarios without alternative tokens.
>
> 3) Multiple Target and Alternative Tokens: TDD accommodates scenarios involving multiple target and alternative tokens by aggregating their probabilities in Equations 2 and 4. The effectiveness of TDD in these settings was assessed through experiments using AG’s News and SST2 datasets, with details provided in the Appendix B.2.3 of our revised paper. These results in Table 6 demonstrate TDD’s effectiveness for multiple target and alternative tokens.
>
> 4) Generated Sequence Attribute Control: The experiments described in Section 5, focusing on toxic language suppression and sentiment steering, further confirm TDD's applicability in controlling attributes of generated sequences.
>
> More details are available in Section 3.6 and Appendix B.2 in our revised manuscript.
>
> **Defined Target and Alternative Tokens in Section 5**
>
> For zero-shot toxic language suppression, we utilized a predefined list of toxic words from WORDFILTER as target tokens, treating all other tokens as alternatives. In this context, considering all other tokens as alternatives is equivalent to having no alternatives.
> For zero-shot positive sentiment steering, we identified negative words from SenticNet as target tokens, using positive words as alternative tokens. This approach allows TDD to identify key words that could lead to negative outcomes and suppress them. To generate negative responses, we simply reverse the roles of the target and alternative tokens, while keeping other settings constant.

---

> > ### Author Response · Authors · 2023-11-16
> > **Response to Reviewer jvM3 (2/2)**
> >
> > [**W2**] The analyses presented in this paper are not really causal analyses, despite claims that the LM head “elucidate causal relationships between prompts and LLM outputs” (page 4). No controlled variable is specified, and no treatment effect is measured. This paper could be much stronger if some sort of causal scores are computed, for example, how large is the treatment effect of changing w to w_a causes the toxicity of the generated texts. Currently, the numbers shown in Tables 3 and 4 look similar to the treatment effects, but without more descriptions about how the variables are controlled, we can’t really say the numbers are treatment effects.
> >
> > [**A3**] We appreciate your insightful feedback. In response, we have expanded our causal analysis by conducting three distinct experiments and analyses to assess treatment effects: 1) Swapping target and alternative tokens; 2) Randomly assigning importance to each input token; 3) Employing varied explanation methods while other operations remain the same. These strategies are designed to confirm that the superior generation-control performance of TDD primarily stems from its ability to effectively identify token saliency for explanations, rather than from other extraneous factors, such as the mere substitution of an input token with a space token.
> >
> > Table 10 in Appendix K presents results for toxic analyses utilizing these three strategies.  Table 11 shows the results of sentiment steering using the three strategies. The token swapping (TDD-CTA), random saliency allocation (TDD-RA) and other explanation methods including Con-GI and Con-GN are significantly outperformed by TDD. These results verify that the superior generation-control performance of TDD primarily stems from its ability to effectively identify token saliency for explanations, rather than from other extraneous factors.
> >
> > The revisions and details are available in Appendix K in our revised paper.
> >
> > [**W3**] A minor point about writing styles: Excessive adverbs (e.g., those in the sentence “elegantly simple yet remarkably effective”) can make the paper read less rigorous, rendering the text resemble more of a social media post than a scientific paper.
> >
> > [**A4**] Thank you for highlighting this issue. We have meticulously revised the paper to minimize excessive adverbs, ensuring rigor.

---

> > > ### Comment · Reviewer_jvM3 · 2023-11-22
> > > **Reviewer reply**
> > >
> > > Thank you for the response and the updates to the pdf. These address most of my concerns, especially about the causal analysis. Clarifying the scope of TDD is helpful as well. I'm happy to raise my score.

---

> > > > ### Author Response · Authors · 2023-11-22
> > > > **Response to Reviewer jvM3**
> > > >
> > > > Dear Reviewer jvM3,
> > > >
> > > > Thank you very much for your follow-up and the positive feedback. We are pleased to know that most of your concerns have been successfully addressed.

---

### Official Review · Reviewer_1T7y · 2023-10-31

**Soundness:** 4 excellent
**Presentation:** 3 good
**Contribution:** 3 good
**Rating:** 8
**Confidence:** 4

**Summary:**

The paper introduces a novel method for constrastive XAI for autoregressive LMs. Instead of using gradients or attention maps, the introduced TDD approach is based on token distributions. TDD aims to explain token influence in the input prompt. To this end, three variants are presented: forward, backward, and bidirectional, as the authors describe, each offering unique insights into token relevance.
The introduced approach is evaluated based on the explanations' faithfulness and the capability to steer the model’s outcome.
Steering is done by replacing the most influential tokens and replacing them, e.g., with white space tokens.

**Strengths:**

- The proposed method is a simple and efficient method
- In general, the paper is well-written, with a clear introduction method and description of experiments
- Evaluation of multiple autoregressive models
- The authors seem to provide all the necessary details for reproducible experiments
- Additional showcasing on interesting and important use cases

**Weaknesses:**

- The difference between the TDD variants is not well discussed. While the applications are very interesting, the authors could have used the space to elaborate on the differences between the introduced variants.
- Captions Table 1 and Table 3 are too sparse.
- While not being a contrastive XAI method, important related work on explainability for autoregressive LMs missing:
ATMAN: Understanding Transformer Predictions Through Memory Efficient Attention Manipulation. Björn Deiseroth, Mayukh Deb, Samuel Weinbach, Manuel Brack, Patrick Schramowski, Kristian Kersting. In Proceedings of NeurIPS 2023

**Questions:**

- Can you elaborate on the difference between TDD and AtMan [Deiseroth et al., 2023] (see weaknesses) beyond contrastive explanations? It seems to be more similar to TDD than Rollout while at the same time relying on attention layers.
- In the steering demonstration, multiple alternative tokens are used at the same time. As far as I understood, in Sec. 4, only one alternative token is selected. How robust is TDD when multiple alternative tokens are used?
- Beyond the benchmark experiments, how would one select an alternative token to explain the LMs decision?
- Which TDD variant is applied in the steering experiments? You mentioned that each TDD variant offers unique insights into token relevance. Can you further elaborate on this regard? Unfortunately, I could not find a discussion in the paper. Are there scenarios where one variant is beneficial? 

---

> ### Author Response · Authors · 2023-11-16
> **Response to Reviewer 1T7y  (1/2)**
>
> Dear Reviewer 1T7y,
>
> Thank you for your thorough review and insightful feedback on our paper. We have addressed each identified weakness (**W**) and responded to the questions (**Q**) raised. Our detailed responses (**A**) are summarized below.
>
> [**Q1**] Can you elaborate on the difference between TDD and AtMan [Deiseroth et al., 2023] (see weaknesses) beyond contrastive explanations?
>
> [**A1**] Thank you for your insightful feedback. We apologize for the oversight regarding the omitted reference and have addressed this in our revised manuscript. To clarify, the primary distinctions between our TDD method and the AtMan approach are as follows:
>
> 1. Saliency Estimation Medium: AtMan evaluates input saliency by iteratively manipulating the attentions of each input token and observing changes in cross-entropy. In contrast, TDD utilizes the token distribution as the medium for calculating token saliency.
>
> 2. Inference Process: AtMan introduces noise at the beginning of forward propagation, whereas TDD performs inference directly at the end of propagation, after the forward propagation process is complete.
>
> 3. Parameter Dependence: TDD has the advantage of not requiring any hyperparameters. On the other hand, AtMan necessitates setting hyperparameters, such as the suppression factor and the lower bound of cosine similarity, which can vary across different datasets.
>
> The revision can be found in Section 2.1 in our revised paper.
>
>
> [**Q2**] How robust is TDD when multiple alternative tokens are used?
>
> [**A2**] Thanks for your professional comment. To affirm the robustness of our TDD method, we expanded our experimental scope to include additional datasets featuring multiple target and alternative tokens. These experiments demonstrate TDD's superior performance over strong baseline methods across diverse datasets. Further details are provided below.
>
> Experiment Setup: In the absence of ready-made datasets for evaluating multiple alternative tokens, we leverage a prompt-learning framework to simulate such an environment. We select two datasets: AG’s News for topic classification and SST2 for sentiment analysis. The inputs are structured as cloze questions, with AG’s News framed as "<input> This topic is about___" and SST2 as "<input> It was __". We incorporate multiple target and alternative tokens by utilizing label words from the KPT method. For each sample, the label words corresponding to the ground-truth label serve as target tokens, while those from other classes are alternative tokens.
>
> Table 6 in Appendix B.2.3 presents the performance comparison of TDD-bidirectional against other baselines. In the SST2 dataset, TDD surpasses SOTA methods by around 7% in AOPC and demonstrates a 6% enhancement in Suff over current baselines. In the AG's News dataset, TDD achieves a 3%-5% margin over existing methods in both AOPC and Suff. These findings verify TDD's efficacy in scenarios involving multiple target and alternative tokens.
>
> More details can be found in Section 3.6 and Appendix B.2.3 in our revised manuscript.
>
> [**Q3**] Beyond the benchmark experiments, how would one select an alternative token to explain the LMs decision?
>
> [**A3**] Thank you for highlighting this aspect. In response to your query, we approach the selection of alternative tokens in three distinct ways:
>
> 1. Omitting Alternative Tokens: Alternative tokens can be excluded from explanations. By assigning a probability of zero to alternative tokens in Equations 2 and 4, we negate their necessity. Our experiments in Appendix B.2.2, which involved the omission of alternative tokens, confirm the effectiveness of our TDD method.
>
> 2. Choosing Based on Linguistic Features: The selection of alternative tokens can align with specific linguistic features under study. For instance, to understand why the model generates "waitresses" in response to the prompt "Amanda was respected by some", we can vary the alternative tokens. For syntactic and morphological analysis, an alternative like "waitress" may be used, while "pictures" may be chosen for semantic analysis. This also underscores our preference for contrastive explanation over conventional explanations in TDD, which we find offers quantifiably superior insights into grammatical phenomena and enhances model simulatability for users.
>
> 3. Observing LLM Output Probabilities: A practical method involves examining the LLM's top-k or top-p output tokens. This approach is straightforward and effective for explaining the model’s choices among similar or commonly confused tokens.

---

> ### Author Response · Authors · 2023-11-16
> **Response to Reviewer 1T7y  (2/2)**
>
> [**Q4**] Which TDD variant is applied in the steering experiments? You mentioned that each TDD variant offers unique insights into token relevance. Can you further elaborate on this regard? Unfortunately, I could not find a discussion in the paper. Are there scenarios where one variant is beneficial?
>
> [**A4**] We appreciate your feedback and have incorporated detailed discussions about the TDD variants and its use in our steering experiments.
>
> 1. In our steering experiments, we specifically chose the TDD-bidirectional variant, as detailed in the revised manuscript. This decision was based on its superior performance observed in our main experiments, which are thoroughly discussed in Section 4.
>
> **2. Detailed Variant Discussion.**
>
> a. Calculation Direction: The TDD-forward evaluates token importance sequentially from the first to the last input token. Conversely, TDD-backward assesses saliency in reverse, from the last input token to the first. TDD-bidirectional, drawing inspiration from bidirectional neural networks, combines saliency estimates from both TDD-forward and TDD-backward.
>
> b. Information Perspective: TDD-backward is less influenced by linguistic conventions, offering a more targeted approach. The TDD-forward overlooks the inherent linguistic influence of individual tokens. Consider the sentence completion by the LLM: “ This design is quite novel and fantastic. I really ___”, our aim is to explain why the LLM generates “like” instead of “hate”. Linguistic conventions render it improbable to predict tokens such as “like” or “hate” following “is” or “novel”, indicating potential inaccuracies in token contribution assessments. TDD-backward can mitigate this issue. Specifically, in TDD-backward, the words “I really ___” are fed to the LLM in the first two iterations. Subsequently, other tokens, such as “fantastic”, are progressively introduced This initial phase sharpens the model's focus, enhancing its accuracy in predicting words such as “like” or “hate” as the LLM consistently generates predictions following the phrase "… I really __" in each iteration.
>
> **3. Application Scenarios** TDD-forward is preferable in time-sensitive or computationally constrained scenarios, as it requires only one forward propagation due to the auto-regressive structure of LLMs. TDD-backward and TDD-bidirectional are better suited for contexts where time is not a constraint, and there is a higher demand for explanation fidelity. They demand iterations equal to the input length for saliency estimation. The revised paper’s Table 2 showcases the higher faithfulness of explanations by TDD-backward and TDD-bidirectional, while Table 9 confirms the computational efficiency of TDD-forward.
>
> More details are provided in Appendix B.1 and G in our revised paper.
>
> [**W1**] The difference between the TDD variants is not well discussed. While the applications are very interesting, the authors could have used the space to elaborate on the differences between the introduced variants.
>
> [**A5**] Please refer to our response [A4] to question 4 [Q4].
>
> [**W2**] Captions Table 1 and Table 3 are too sparse.
>
> [**A6**] Thanks for pointing this out. We have revised the captions for Tables 1, 3 and 4 accordingly in our modified manuscript.
>
> [**W3**] While not being a contrastive XAI method, important related work on explainability for autoregressive LMs missing: ATMAN: Understanding Transformer Predictions Through Memory Efficient Attention Manipulation. Björn Deiseroth, Mayukh Deb, Samuel Weinbach, Manuel Brack, Patrick Schramowski, Kristian Kersting. In Proceedings of NeurIPS 2023
>
> [**A7**] Please refer to our response [A1] to question 1 [Q1].

---

> > ### Comment · Reviewer_1T7y · 2023-11-22
> >
> > Thank you for the clarifications and for addressing the remaining concerns. I'm happy to keep my score and vote to accept the paper.

---

> > > ### Author Response · Authors · 2023-11-23
> > > **Response to Reviewer 1T7y**
> > >
> > > Dear Reviewer 1T7y,
> > >
> > > We greatly appreciate your positive feedback. It is encouraging to know that we have addressed most of your concerns effectively.

---

### Official Review · Reviewer_4Y86 · 2023-10-31

**Soundness:** 3 good
**Presentation:** 3 good
**Contribution:** 3 good
**Rating:** 6
**Confidence:** 3

**Summary:**

This paper presents a new method via using Token Distribution Dynamics (called TDD) to both interpret and control LLMs' generations.

The authors compared the proposed method with a few baselines (attention rollout, contrastive gradient) and show TDD obtains a more precise explanation with lower sufficiency. The authors also put TDD into real use for toxic language suppression and sentiment steering, and show TDD can effectively control LLMs' outputs.

**Strengths:**

- The idea of analyzing the token distributions throughout the progression of prediction is quite interesting. The idea is simple but seems to be quite useful in unveiling the importance of input tokens when providing contrastive explanations.

- The authors did experiments over a fairly comprehensive set of language models including GPT-2/J, BLOOM, and LLaMA.

- The applications on toxic language suppression and sentiment steering further demonstrate the usefulness of the proposed TDD method.

**Weaknesses:**

- One simple way to control LLMs' generations is via prompting for style transfer, e.g., ask the model to transform the outputs into "less toxic" content, or "positive/negative" sentiment. How would this baseline compare to the proposed TDD?

- The models used in experiments are relatively smaller models (maximum size 7B), how would the proposed approach work on larger models (e.g., llama-2 13B, 70B)? What is the computation cost (efficiency, memory) for running TDD over larger models?

- The proposed TDD provides contrastive explanations on the token level, it would be interesting to see if this method can be extended to higher-level concepts, e.g., why sometimes a model chooses to generate an unfaithful output.

**Questions:**

- Could the authors add a simple baseline by prompting the LLMs for style transfer on the outputs?

- What is the computation cost (efficiency, memory) for running TDD over larger models?

---

> ### Author Response · Authors · 2023-11-16
> **Response to Reviewer 4Y86**
>
> Dear Reviewer 4Y86,
>
> Thank you for taking the time to review our paper and providing insightful feedback. Our responses (**A**) to the mentioned weaknesses (**W**) and posed questions (**Q**) are summarized as follows.
>
> [**Q1 & W1**]: One simple way to control LLMs' generations is via prompting for style transfer. How would this baseline compare to the proposed TDD? Could the authors add a simple baseline by prompting the LLMs for style transfer on the outputs?
>
> [**A1**]: We appreciate the valuable suggestion offered. In response, we have incorporated two robust baselines from the field of text style transfer aimed at regulating the toxicity and sentiment in LLM outputs.
>
> We compare TDD with two baselines from this field. Style Prompting (SP) (Reif et al., 2022) integrates specific style instructions like toxicity into prompts, while Augmented Style Prompting (ASP) (Reif et al., 2022) enhances this approach by introducing varied rewriting examples for a broader application. Detailed information regarding these two prompting strategies can be found in Appendix H and M.
>
> In our revised manuscript, Tables 3 and 4 detail the experimental results. SP records a toxicity of 0.47, ASP attains 0.41, whereas TDD achieves a notably lower score of 0.20, significantly outperforming SP and ASP. In terms of negative sentiment steering, TDD surpasses both SP and ASP by a substantial margin of about 0.30. For positive sentiment steering, TDD exceeds SP and ASP by more than 0.20, underscoring its considerable advantage over style transfer methods.
>
>
>
> [**Q2&W2**]:
> How would the proposed approach work on larger models (e.g., llama-2 13B, 70B)? What is the computation cost (efficiency, memory) for running TDD over larger models?
>
> [**A2**]: Thank you for your insightful feedback. In response, we have expanded our research to include experiments on explanation faithfulness using larger models such as LLaMA2-13B and OPT-30B to ensure its scalability. Additionally, we have included a comparison of the computational costs for both our methods and the baseline approaches. The key findings are summarized below.
>
> **Scalability**. We conduct experiments with LLaMA2-13B and OPT-30B (Zhang et al., 2022) to assess the effectiveness of TDD in explaining larger models. The summarized results in Table 8 in the revised manuscript reveal that TDD-forward outperforms the baselines by margins of 3.15% in AOPC and 4.4% in Suff using LLaMA2-13B. Both TDD-backward and TDD-bidirectional demonstrate superior performance, exceeding the baselines by more than 4% in AOPC and over 5% in Suff. For OPT-30B, TDD surpasses competitive baselines by a significant margin of 4% to 6%. These results prove TDD's scalability and effectiveness in larger models.
>
> **Computation Cost**. For the computational cost analysis, we evaluate the average memory usage and processing time required by our method for processing a single input sample. Consistency is maintained across all experiments, which are conducted on an NVIDIA RTX A5000 GPU. For models larger than 6 billion parameters, their 4-bit versions are utilized. Detailed memory and time metrics are presented in Table 9 in the paper. Regarding memory consumption, Rollout and the three TDD variants (TDD-forward, TDD-backward, and TDD-bidirectional) are the most efficient. In terms of processing time, TDD-forward and Rollout emerge as the fastest, whereas TDD-backward and TDD-bidirectional exhibit slightly longer processing time.
>
> More details are provided in Section 4.6 and Appendix F, G in our revised manuscript.
>
> [**W3**]: The proposed TDD provides contrastive explanations on the token level, it would be interesting to see if this method can be extended to higher-level concepts, e.g., why sometimes a model chooses to generate an unfaithful output.
>
> [**A3**]: Thank you for your constructive suggestions. We agree that applying our method to higher-level concepts is an interesting direction for future work.
> Generally speaking, LLM generations can be explained from basic to advanced levels. At the token level, it's crucial to understand and explain the reasons behind the generation of specific tokens. The sequence-attribute level involves interpreting why LLMs produce sequences with attributes like toxicity or positivity. The third, more complex level, examines the production of biased or unfaithful outputs. We've demonstrated our method's effectiveness at the first and second levels through experiments in Section 4 and controlled text generation in Section 5. Applying our approach to the third level remains a goal for future research, aiming to deepen our understanding of LLMs and further their development.
>
>
> References
>
> Reif, E., Ippolito, D., Yuan, A., Coenen, A., Callison-Burch, C., & Wei, J. (2022). A Recipe for Arbitrary Text Style Transfer with Large Language Models.
>
> Zhang S, Roller S, Goyal N, et al. Opt: Open pre-trained transformer language models[J].

---

> > ### Comment · Reviewer_4Y86 · 2023-11-22
> > **Thanks for the response**
> >
> > Thanks the authors for adding the details. Overall I think this is an interesting paper and lean towards acceptance.
> >
> > For SP and ASP though, I'm surprised that they're not effective in either task (very little difference compared to the baseline GPT-2 model). Have you studied why this is the case? Is it because of how the prompt is written (e.g., instead of "less toxic", we can say "not toxic at all", can this lead to more toxicity suppression?), or because the model doesn't follow the instruction very well (e.g., using an instruction-tuned model would work better)?

---

> ### Author Response · Authors · 2023-11-22
> **Response to Reviewer 4Y86**
>
> Dear Reviewer 4Y86,
>
> Thank you very much for your follow-up and positive assessment. We are pleased to note that our responses have addressed most of your concerns.
>
> Regarding the main reason why SP and ASP are less effective, we argue it is due to GPT2's limited capacity to comprehend and adhere to instructions. Our tests involved various prompt styles. For example, to generate positive outputs, we experimented with prompts containing keywords such as "be more positive/good/very positive/totally positive" and "be less negative/bad," including synonyms of these terms. Unfortunately, these prompts proved largely ineffective. The most successful prompts have been carefully selected and are detailed in our revised paper. Consequently, we infer that GPT2 struggles with following SP and ASP instructions, even though some demonstrations are included in ASP.
>
> We would also like to highlight that in the original SP and ASP paper, their effectiveness was demonstrated using much larger models like LaMDA-137B and GPT3-175B, as opposed to the significantly smaller GPT2-774M and LlaMA-7B models. This leads us to believe that SP and ASP techniques may be more suited to and effective with larger or instruction-tuned models.
>
> Thank you again for your response. Please let us know if you have further questions.

---

### Author Response · Authors · 2023-11-16
**Summary**

Dear Reviewers,

We deeply appreciate your insightful feedback and questions. Our team is encouraged by your recognition of our work as a novel/ interesting approach for unveiling and controlling prompt influence in LLMs. Following your suggestions, we have thoroughly revised the manuscript, highlighting all modifications in blue. Key revisions include:

•	Scope Clarification: Our proposed TDD is suitable for situations involving single or multiple target tokens, with or without alternative tokens. We also showcase the application of TDD in zero-shot controlled text generation, addressing a challenge frequently encountered in real-world scenarios.

•	Experimental Enhancements: In line with your recommendations, we've conducted additional experiments to provide a more comprehensive and causal understanding.

•	Improved Explanations: We have addressed and clarified parts of the paper that were previously unclear, as indicated in your feedback.

•	Expanded Literature Review: The paper now includes a more extensive comparison and analysis of related literature.

•	Writing Refinement: The entire paper has been meticulously reviewed and revised to ensure scientific accuracy and clarity.

The details of revisions can be found in our official responses and revised paper. Please do not hesitate to contact us for any further suggestions or discussions.

With Gratitude,

Authors of Paper3427

---

### Meta-Review · Area_Chair_LMiS · 2023-12-09

**Metareview:**

This paper proposes a new method for explaining autoregressive LM generation in a contrastive way. Instead of using gradients or attention maps, the new approach is based on token distributions and aims to explain the influence of tokens  in the input prompt. Three variants are presented: forward, backward, and bidirectional. Each offering unique insights into token relevance. The introduced approach is evaluated based on the explanations' faithfulness and the capability to steer the model’s outcome. Steering is done by replacing the most influential tokens and replacing them, e.g., with white space tokens. The idea of analyzing the token distributions throughout the progression of prediction is simple and interesting. The authors evaluated on a wide range of models (GPT-2/J, BLOOM, and LLaMA). On the other hand, for the studied problems like sentiment manipulation and detoxified generation, it'd be interesting to see how instruction-finetuned LLMs will perform (e.g., instruction-finetuned LLaMa 7B and/or 13B), evaluated with human annotations (as automatic evaluation is often unreliable especially LLMs may not strictly follow the desired output format).

**Justification For Why Not Higher Score:**

More (human) evaluation and analyses of instruction-finetuned LLMs on the studied tasks (sentiment/toxicity manipulation) are desirable.

**Justification For Why Not Lower Score:**

- interesting simple method for constructing contrastive explanation of prompt token influence
- fairly comprehensive experiments

---

### Decision · Program_Chairs · 2024-01-16

Accept (poster)